# UNIFIED PARAMETER-EFFICIENT UNLEARNING FOR LLMS

**Chenlu Ding**[1][*][†]     **Jiancan Wu**[1][†][‡]     **Yancheng Yuan**[2][‡]     **Jinda Lu**[1]

**Kai Zhang**[1]     **Alex Su**[1]     **Xiang Wang**[1]     **Xiangnan He**[13][‡]

[1] University of Science and Technology of China     [2] Hong Kong Polytechnic University
[3] MoE Key Laboratory of Brain-inspired Intelligent Perception and Cognition, USTC

## ABSTRACT

The advent of Large Language Models (LLMs) has revolutionized natural language processing, enabling advanced understanding and reasoning capabilities across a variety of tasks. Fine-tuning these models for specific domains, particularly through Parameter-Efficient Fine-Tuning (PEFT) strategies like LoRA, has become a prevalent practice due to its efficiency. However, this raises significant privacy and security concerns, as models may inadvertently retain and disseminate sensitive or undesirable information. To address these issues, we introduce a novel instance-wise unlearning framework, LLMEraser, which systematically categorizes unlearning tasks and applies precise parameter adjustments using influence functions. Unlike traditional unlearning techniques that are often limited in scope and require extensive retraining, LLMEraser is designed to handle a broad spectrum of unlearning tasks without compromising model performance. Extensive experiments on benchmark datasets demonstrate that LLMEraser excels in efficiently managing various unlearning scenarios while maintaining the overall integrity and efficacy of the models. Our code is available at https://github.com/oceanoceanna/LLMEraser.

## 1 INTRODUCTION

Large language models (LLMs) demonstrate remarkable capabilities in knowledge understanding and complex reasoning (Li et al., 2023; Zhang et al., 2024b; Li, 2024; Li et al., 2024; Lee et al., 2024), having sparked increasing interest in adapting LLMs to specific domains through fine-tuning techniques (Li & Liang, 2021; Dettmers et al., 2023; Zhang et al., 2023; Zaken et al., 2022). Among them, Parameter-Efficient Fine-Tuning (PEFT) (Li & Liang, 2021; Liu et al., 2021), such as LoRA (Hu et al., 2022), has emerged as the mainstream paradigm, offering significant reductions in resource costs by fine-tuning only a small subset of parameters. While highly effective, the reliance on domain-specific data for fine-tuning raises concerns regarding data leakage and privacy (Lu et al., 2024; Blanco-Justicia et al., 2024), such as potentially memorizing or propagating sensitive, biased, copyrighted, or harmful information (Liu et al., 2024c; Qu et al., 2024). In this light, researchers have introduced unlearning techniques (Jang et al., 2023; Kurmanji et al., 2023; Kumar et al., 2023) into LLMs, to "forget" specific data without requiring the time-consuming and resource-intensive process of retraining.

Prior efforts in exploring unlearning in LLMs primarily focus on removing specific concepts (Kassem et al., 2023; Jang et al., 2023). A typical example is the erasure of LLM's ability to recall information related to the Harry Potter series (Eldan & Russinovich, 2023). While these efforts yield valuable insights, they risk inadvertently affecting related concepts, such as other novels with similar titles. In this work, we broaden the scope by investigating instance-wise unlearning

---

[*] Work done at The Hong Kong Polytechnic University.

[†] Equal contribution. {*dingchenlu200103,wujcan*}*@gmail.com*

[‡] Correspondence to Jiancan Wu, Yancheng Yuan, and Xiangnan He. {*wujcan@gmail.com, yancheng.yuan@polyu.edu.hk, xiangnanhe@gmail.com*}

Table 1: A summary of existing LLM unlearning methods and their application scenarios. 'E' and 'A' are abbreviations for Exact unlearning and Approximate unlearning, respectively.

| Related Work | Mode | Method | Preserve Model Architecture | Free from Retrain/Pretrain | IR | QM | RC |
|---|---|---|---|---|---|---|---|
| Retrain | - | Retrain | ✓ | ✗ | ✓ | ✓ | ✓ |
| SISA (Bourtoule et al., 2021) | E | Retrain Sub-model | ✗ | ✗ | ✓ | ✓ | ✓ |
| FairSISA (Kadhe et al., 2023) | E | Retrain Sub-model | ✗ | ✗ | ✓ | ✓ | ✓ |
| APA (Hu et al., 2024c) | E | Retrain Sub-model | ✗ | ✗ | ✓ | ✓ | ✓ |
| Gradient Ascent | A | Fine-tuning | ✗ | ✓ | ✓ | ✗ | ✗ |
| EUL (Chen & Yang, 2023) | A | Fine-tuning | ✗ | ✓ | ✓ | ✗ | ✗ |
| E2URec (Wang et al., 2024) | A | Fine-tuning | ✗ | ✗ | ✓ | ✗ | ✗ |
| LLMEraser (Ours) | A | Parameter Editing | ✓ | ✓ | ✓ | ✓ | ✓ |

tasks, which allow us to target more nuanced aspects of model behavior. To this end, we first present various instance-wise unlearning tasks for LLMs, as illustrated in Figure 1. More case studies can be found in Appendix G. Specifically, consider a training instance $z = (x, y)$ in a supervised fine-tuning dataset, where $x$ represents the query and $y$ is the response. We can categorize the LLMs unlearning tasks at the instance level as follows:

- **Instance Removal (IR).** It removes the sample $z = (x, y)$ from the training set.
- **Query Modification (QM).** It adjusts the input tokens in query $x$, such as removing specific noisy tokens or correcting certain erroneous tokens.
- **Response Correction (RC).** It corrects the model's response $y$, including updating outdated answers or rectifying incorrect classification results.

In this work, we focus on unlearning the domain-specific data used solely in PEFT, which requires updating the PEFT adapters (*e.g.,* LoRA). Technically, recent LLM-unlearning efforts can be roughly grouped into two categories. **Exact unlearning** approaches divide data into disjoint shards and retrain adapters (Bourtoule et al., 2021; Hu et al., 2024c). Despite effectiveness, these methods have inherent limitations — inevitably destroying the model's original structure and necessitating the retraining cost. **Approximate unlearning** methods, on the other hand, aim to replicate the performance of the retrained model, often aligning the output of the target data closely with randomness through KL-divergence-based PEFT (Liu et al., 2024a; Qu et al., 2024). Nonetheless, this paradigm primarily focuses on data removal (*e.g.,* IR) and hardly corrects biased or inaccurate data (*e.g.,* QM, RC), as it falls short in guiding the output of the target data towards accurate information, rather than mere randomness. See Table 1 for the summary of current LLMs unlearning methods, with detailed descriptions available in Appendix A. Overall, both approaches struggle to efficiently handle these instance-wise LLM unlearning tasks and are not specifically designed for unlearning within the PEFT framework. It calls for a general LLM unlearning method capable of addressing these various tasks.

In pursuit of parameter-efficient unlearning, we identify the influence function (Koh & Liang, 2017) as a promising tool. At its core is to formulate the parameter changes caused by perturbations in the form of the inverse Hessian-vector-product (Agarwal et al., 2016), where Hessian matrix represents the curvature of the loss function *w.r.t.* model parameters. However, the direct application of the influence function to LLMs presents two significant challenges: the expensive cost of calculating the inverse Hessian-vector-product for vast model parameters and the cumulative errors introduced by approximation strategies (*e.g.,* stochastic estimation (Agarwal et al., 2016)). Consequently, the use of influence functions for LLM unlearning remains largely underexplored. To fill this research gap, we propose a unified parameter-efficient unlearning framework, LLMEraser, for various instance-wise unlearning tasks. Specifically, for each type of unlearning task, LLMEraser leverages influence functions to directly calculate the parameter changes in the PEFT adapters and then efficiently update the adapter parameters, thus bypassing the need for time-consuming model retraining or fine-tuning. Furthermore, we reformulate the calculation of the inverse Hessian-vector-product into a finite-sum quadratic programming problem (Nesterov, 2013; Beck & Teboulle, 2009), significantly reducing computational complexity while mitigating the approximation errors from stochastic estimation. LLMEraser has several advantages: model-agnostic, applicable to various instance-wise unlearning tasks, and ensuring fast model updates. We conduct experiments on both LLMs and Multimodal Large Language Models (MLLMs), specifically focusing on LLMs for Recommenda-

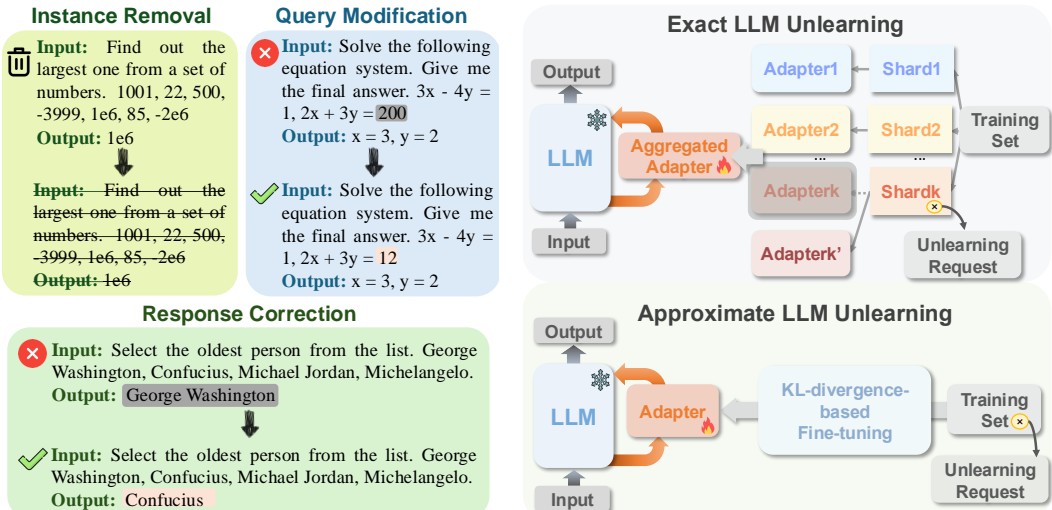

(a) Taxonomy of LLM unlearning tasks.  (b) Overview of exact/approximate LLM Unlearning.

Figure 1: 1a: A brief description of the different types of LLM unlearning tasks. 1b: The framework of exact LLM unlearning method, approximate unlearning method.

tion (LLM4Rec) as well as MLLM relation mining tasks to validate the effectiveness of LLMEraser. Our extensive evaluations across these diverse scenarios demonstrate that LLMEraser consistently outperforms the state-of-the-art unlearning methods.

## 2 PRELIMINARY

This section introduces key concepts underpinning our methodology. We cover instruction tuning to enhance LLMs' understanding of human instructions, followed by PEFT, highlighting LoRA for efficient updates. Lastly, we discuss the influence function, which analyzes parameter changes from data perturbations. These foundations set the stage for the techniques discussed later.

### 2.1 INSTRUCTION TUNING

Instruction tuning is a key technique that leverages carefully curated datasets of human-annotated instructions and corresponding responses to enhance LLMs' capacity to comprehend and respond to human instructions (Wei et al., 2022; Liu et al., 2023b; Sanh et al., 2022). Given a downstream task dataset $\mathcal{Z} = \{z|z = (x,y)\}$ containing $n$ instances, where $x$ represents a description of the human instruction and $y$ is the corresponding response, LLMs are fine-tuned using the following autoregressive (Brown et al., 2020; Touvron et al., 2023a) objective:

$$\max_{\Phi} \sum_{(x,y)\in\mathcal{Z}} \sum_{t=1}^{|y|} \log\left(P\left(y_t \mid x, y_{<t}; \Phi\right)\right), \qquad (1)$$

where $\Phi$ is LLMs' parameters, $y_t$ is the $t$-th token of $y$, and $y_{<t}$ represents tokens preceding $y_t$.

### 2.2 PARAMETER-EFFICIENT FINE-TUNING

LLMs typically consist of billions of parameters, making full fine-tuning computationally expensive. Parameter-Efficient Fine-Tuning (PEFT) addresses this challenge by updating only a small number of the parameters while still achieving satisfactory performance. Among them, LoRA (Hu et al., 2022) stands out as particularly effective, which freezes the original pretrained parameters while introducing pairs of low-rank-decomposition weight matrices to simulate parameter updates. Formally, the optimization objective for LoRA is expressed as follows:

$$\max_{\Theta} \sum_{(x,y)\in\mathcal{Z}} \sum_{t=1}^{|y|} \log\left(P\left(y_t \mid x, y_{<t}; \Phi + \Delta\Phi(\Theta)\right)\right), \qquad (2)$$

where $\Theta$ is the trainable parameters that is significantly smaller in size compared to $\Phi$.

## 2.3 INFLUENCE FUNCTION

The influence function was first applied in machine learning by Koh & Liang (2017) to analyze the outputs of black-box models. For the dataset $\mathcal{Z}$, we focus on the following empirical risk minimization (Shalev-Shwartz & Ben-David, 2014; Vapnik, 1998; Bartlett & Mendelson, 2002) problem:

$$\hat{\Theta} \in \arg\min_{\Theta} \left\{ R(\mathcal{Z};\Theta) | R(\mathcal{Z};\Theta) := \frac{1}{n} \sum_{(x,y) \in \mathcal{Z}} \mathcal{L}\left((x,y);\Theta\right) \right\}, \tag{3}$$

where $\Theta$ is the trainable model parameter and $\hat{\Theta}$ is the minimizer of Equation 3. $\mathcal{L}(\cdot;\Theta)$ is the loss function, and for Equation 2, it is defined as:

$$\mathcal{L}\left((x,y);\Theta\right) = -\sum_{t=1}^{|y|} \log\left(P\left(y_t \mid x, y_{<t}; \Phi + \Delta\Phi(\Theta)\right)\right). \tag{4}$$

When a training example $(x,y)$ is upweighted by an infinitesimal amount $\epsilon$, the perturbed loss for $\hat{\Theta}_{\text{new}}(\epsilon)$ can be expressed as:

$$\hat{\Theta}_{\text{new}}(\epsilon) \in \arg\min_{\Theta} \left\{ \widehat{\mathcal{L}}\left(\mathcal{Z}, (x,y), \epsilon; \Theta\right) | \widehat{\mathcal{L}}\left(\mathcal{Z}, (x,y), \epsilon; \Theta\right) := R(\mathcal{Z};\Theta) + \epsilon\mathcal{L}\left((x,y);\Theta\right) \right\}. \tag{5}$$

When $\epsilon \approx 0$, the parameter change $\Delta\Theta(\epsilon) = \hat{\Theta}_{\text{new}}(\epsilon) - \hat{\Theta}$ can be approximately calculated by applying a Taylor expansion of Equation 3. Please refer to (Koh & Liang, 2017) for detailed derivation. Specifically, $\Delta\Theta(\epsilon)$ can be written as:

$$\Delta\Theta(\epsilon) \approx -\epsilon H_{\hat{\Theta}}^{-1} \nabla_{\Theta} \mathcal{L}\left((x,y);\hat{\Theta}\right), \tag{6}$$

where $H_{\hat{\Theta}} = \nabla_{\Theta}^2 R(\mathcal{Z};\hat{\Theta})$ is the Hessian matrix, $\nabla_{\Theta}\mathcal{L}((x,y);\hat{\Theta})$ represents the gradient of $\mathcal{L}$ *w.r.t.* parameters $\Theta$, evaluated at $\hat{\Theta}$.

## 3 METHOD

In this work, we propose LLMEraser, a framework that updates the PEFT adapter parameters to handle various instance-wise unlearning tasks. As shown in Figure 2, our approach leverages the influence function to directly estimate the parameter changes for various unlearning tasks, circumventing the resource-consuming fine-tuning or retraining procedures. Moreover, we present a novel algorithm to accelerate the computation of the inverse Hessian-vector-product in the influence function, enabling its efficient implementations in LLMs. Finally, we summarize how LLMEraser works.

### 3.1 TAXONOMY OF LLM UNLEARNING TASKS

We focus on instance-wise unlearning tasks for LLMs, specifically for PEFT that uses domain-specific data. For an instance $z = (x,y)$, where $x$ represents the query and $y$ is the response, we propose a taxonomy of unlearning tasks based on the operation applied to the target instance.

**Instance Removal (IR).** When a specific instance $z = (x,y)$ is either restricted from use or contains harmful content, it necessitates complete elimination from the training set, along with its associated influence on the model.

**Query Modification (QM).** This category involves modifying the query $x$, transforming $z = (x,y)$ into $z' = (x',y)$. It could not only delete outdated or incorrect tokens in the query $x$, such as noisy interactions from a user's history, but also update erroneous or outdated tokens with correct ones.

**Response Correction (RC).** Here, the focus is on rectifying the output component $y$ of the instance $z$. That is, replacing $z = (x,y)$ with $z' = (x,y')$. For binary classification tasks, such as answering "Yes" or "No", it corrects mislabeled outputs by flipping the labels. For other tasks, such as multi-class classification or question answering, it is applied to rectify inaccurate responses.

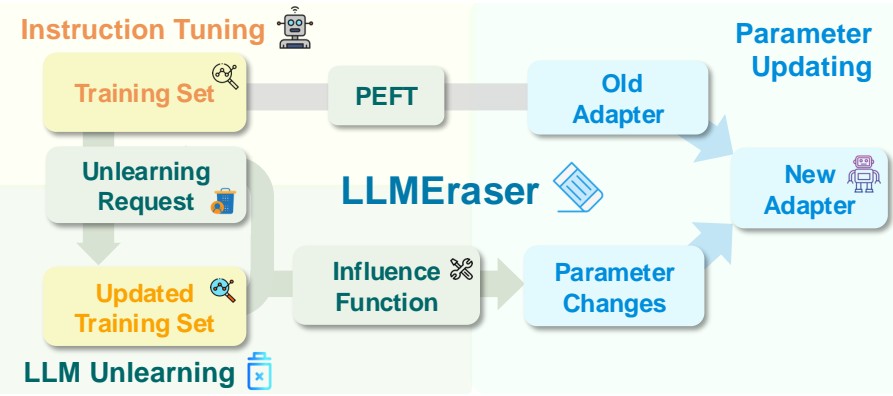

Figure 2: The framework of LLMEraser. The old adapter is obtained through PEFT on domain-specific data. When an unlearning request arrives (*e.g.*, deleting or correcting certain data from the training set), LLMEraser utilizes influence functions to compute the parameter changes caused by such request. These estimated parameter modifications are added to the old adapter's weights, resulting in the new adapter parameters—essentially the unlearned model parameters.

Our proposed taxonomy expands the concept of LLM unlearning beyond the removal of entire instances. It introduces a more fine-grained categorization defined at the token level within both queries and responses, allowing for nuanced control of model behavior.

## 3.2 LLMERASER

The key strength of LLMEraser lies in its capacity to directly estimate the adapter's parameter changes caused by various unlearning tasks. For the sake of clarity and without sacrificing generality, we employ the loss function in LoRA (*cf.* Equation 4) as our example, while other alternatives would yield similar formulations.

To develop a unified approach for solving all unlearning tasks in our taxonomy, we begin by considering a general case where perturbations are applied to both the query ($x$) and response ($y$) components of an instance $z$. This generalized framework allows us to model each specific unlearning task as a special case of this perturbation scenario. Formally, we define the perturbation $\delta$ applied to $z$ as $z_\delta = (x + \delta_x, y + \delta_y)$, where $\delta_x$ and $\delta_y$ represent perturbations to the query and response, respectively. We now formulate the perturbed empirical risk minimization problem as:

$$\widehat{\Theta_\delta}(\epsilon) \in \arg\min_{\Theta} \left\{ R(\mathcal{Z}; \Theta) + \epsilon \mathcal{L}\left((x + \delta_x, y + \delta_y); \Theta\right) - \epsilon \mathcal{L}\left((x, y); \Theta\right) \right\}, \quad (7)$$

where $\widehat{\Theta_\delta}(\epsilon)$ is the minimizer of the optimization problem after applying a perturbation $\delta$ of magnitude $\epsilon$ to the sample $z$. Following the derivation in (Koh & Liang, 2017), when the sample size $n$ is sufficiently large, by taking $\epsilon = \frac{1}{n}$ (*i.e.*, $\epsilon \approx 0$), we can safely estimate the parameter change $\Delta\Theta_\delta$ as follows:

$$\Delta\Theta_\delta \approx \frac{1}{n} \left( \nabla_\Theta^2 R(\mathcal{Z}; \hat{\Theta}) \right)^{-1} \left( \mathcal{G}(x, y) - \mathcal{G}(x + \delta_x, y + \delta_y) \right), \quad (8)$$

where $\mathcal{G}(x, y)$ is an abbreviations for $\nabla_\Theta \mathcal{L}\left((x, y); \hat{\Theta}\right)$. Next, we present the perturbations and corresponding parameter changes for different unlearning tasks.

- **Instance Removal.** The deletion of data corresponds to the perturbation function in Equation 5. By setting $\epsilon = -\frac{1}{n}$ like Equation 6, it is equivalent to removing instance $z$. The set of deleted instances is denoted as $\mathcal{S}_{\text{IR}}$. By aggregating the gradients of all deleted instances, the parameter change $\Delta\Theta_{\text{IR}}$ can be expressed as follows:

$$\Delta\Theta_{\text{IR}} \approx \frac{1}{n} \left( \nabla_\Theta^2 R(\mathcal{Z}; \hat{\Theta}) \right)^{-1} \sum_{(x, y) \in \mathcal{S}_{\text{IR}}} \mathcal{G}(x, y). \quad (9)$$

- **Query Modification**. Modifying certain tokens in the query $x$ is equivalent to perturbing $x$ with $\delta_x$, where $\delta_x$ represents deleting noisy tokens or correcting inaccurate tokens, while keeping the response unchanged (*i.e.*, $\delta_y = 0$). Hence, the perturbed instance $z$ is represented as $z_\delta = (x + \delta_x, y)$,

with the set of instances requiring the removal or modification of specific tokens represented by $\mathcal{S}_{\text{QM}}$. By aggregating the gradients of all instances in $\mathcal{S}_{\text{QM}}$, the parameter change $\Delta\Theta_{\text{QM}}$ induced by query modification can be shown as follows:

$$\Delta\Theta_{\text{QM}} \approx \frac{1}{n}\left(\nabla_{\hat{\Theta}}^2 R(\mathcal{Z};\hat{\Theta})\right)^{-1}\left(\sum_{(x,y)\in\mathcal{S}_{\text{QM}}} \mathcal{G}(x,y) - \sum_{(x+\delta_x,y)\in\mathcal{S}_{\text{QM}}} \nabla_{\Theta}\mathcal{G}(x+\delta_x,y)\right). \quad (10)$$

- **Response Correction.** Correcting the response solely corresponds to $\delta_x = 0$ while perturbing the response $y$ with $\delta_y$. Here $\delta_y$ represents updates to outdated answers or adjustments to erroneous classification results. With $z_\delta = (x, y + \delta_y)$, the set of instances with rectified labels is $\mathcal{S}_{\text{RC}}$. The parameter change $\Delta\Theta_{\text{RC}}$ is as follows:

$$\Delta\Theta_{\text{RC}} \approx \frac{1}{n}\left(\nabla_{\hat{\Theta}}^2 R(\mathcal{Z};\hat{\Theta})\right)^{-1}\left(\sum_{(x,y)\in\mathcal{S}_{\text{RC}}} \mathcal{G}(x,y) - \sum_{(x,y+\delta_y)\in\mathcal{S}_{\text{RC}}} \mathcal{G}(x,y+\delta_y)\right). \quad (11)$$

However, computing inverse Hessian-vector-product results presents significant challenges. Although CG (Hestenes et al., 1952; Fletcher, 2000; Shewchuk et al., 1994) shows some promise, it requires full-batch gradient computation (Koh & Liang, 2017), making it impractical for large-scale datasets. Stochastic estimation (Agarwal et al., 2016) expands $(\nabla_{\hat{\Theta}}^2 R(\mathcal{Z};\hat{\Theta}))^{-1}$ into a truncated power series and iteratively estimates parameter changes, but it suffers from cumulative approximation errors (Blanco-Justicia et al., 2024; Basu et al., 2021). Next, we elaborate a new efficient and scalable algorithm for computing $\Delta\Theta_{\text{Task}}$ for different unlearning tasks.

## 3.3 A New Algorithm for Computing Parameter Changes

Inspired by the previous studies (Ding et al., 2025), LLMEraser reformulates the calculation of parameter changes as solving an equivalent optimization problem expressed in summation form, enabling efficient resolution using mini-batch algorithms. Specifically, we focus on the following optimization problem regarding $\Delta$:

$$\min_{\Delta} F(\Delta) := \frac{1}{2}\Delta^\top \nabla_{\hat{\Theta}}^2 R(\mathcal{Z};\hat{\Theta})\Delta - \langle b, \Delta\rangle, \quad (12)$$

where $\langle,\rangle$ represents the inner product of vectors, and $b$ is defined as:

$$b = \begin{cases} \frac{1}{n}\sum_{(x,y)\in\mathcal{S}_{\text{IR}}} \mathcal{G}(x,y), & \text{if Task = IR} \\ \frac{1}{n}\sum_{(x,y)\in\mathcal{S}_{\text{IM}}} \mathcal{G}(x,y) - \frac{1}{n}\sum_{(x+\delta_x,y)\in\mathcal{S}_{\text{IM}}} \mathcal{G}(x+\delta_x,y), & \text{if Task = IM} \\ \frac{1}{n}\sum_{(x,y)\in\mathcal{S}_{\text{RC}}} \mathcal{G}(x,y) - \frac{1}{n}\sum_{(x,y+\delta_y)\in\mathcal{S}_{\text{RC}}} \mathcal{G}(x,y+\delta_y), & \text{if Task = RC} \end{cases} \quad (13)$$

Since $\hat{\Theta}$ is the minimizer of Equation 3, it satisfies the second-order necessary optimality condition (Nocedal & Wright, 1999; Luenberger et al., 1984; Bertsekas, 1997), resulting in the matrix $\nabla_{\hat{\Theta}}^2 R(\mathcal{Z};\hat{\Theta})$ being symmetric and positive semidefinite. Thus, Equation 12 is essentially a convex quadratic problem, with a gradient of $\nabla_{\hat{\Theta}}^2 R(\mathcal{Z};\hat{\Theta})\Delta - b$.

Given that $\Delta\Theta_{\text{Task}}$ can be interpreted as the solution to the linear system $\nabla_{\hat{\Theta}}^2 R(\mathcal{Z};\hat{\Theta})\Delta = b$, addressing $\Delta\Theta_{\text{Task}}$ is effectively equivalent to optimizing Equation 12. Due to the summation form of $\nabla_{\hat{\Theta}}^2 R(\mathcal{Z};\hat{\Theta})$, Equation 12 can be reformulated as the following finite-sum formation:

$$F(\Delta) = \frac{1}{n}\sum_{(x,y)\in\mathcal{Z}} f\left((x,y),\Delta\right), \quad (14)$$

where $f((x,y),\Delta)$ is defined as:

$$f((x,y),\Delta) = \frac{1}{2}\Delta^\top \nabla_{\hat{\Theta}}^2 \mathcal{L}\left((x,y),\hat{\Theta}\right)\Delta - \langle b, \Delta\rangle. \quad (15)$$

By employing scalable algorithms (*e.g.,* SGD) to optimize problem 12, we can obtain the solution for $\Delta\Theta_{\text{Task}}$. It is worth noting that both the function value and the gradient can be efficiently

computed using the Hessian-vector-product (HVP)[1], reducing the complexity from $\mathcal{O}(p^2)$ to $\mathcal{O}(p)$ (Pearlmutter, 1994), where $p$ is the number of trainable parameters. The pseudocode for computing parameter changes can be found in Appendix B. Error analysis for our proposed algorithm can be found in Appendix D.

### 3.4 THE WORKFLOW OF LLMERASER

LLMEraser focuses on unlearning domain-specific data and updating the parameters of the PEFT adapters. Overall, the workflow of LLMEraser is as follows:

- Leverage domain-specific data and apply PEFT techniques to train and obtain the initial adapter, which captures the model's performance on the original dataset.
- When certain data becomes unavailable, process and validate the unlearning request to ensure compliance with regulations or organizational policies before initiating the unlearning procedure.
- Utilize LLMEraser, which employs influence functions to efficiently calculate the necessary changes in the model parameters resulting from the specified unlearning request. This step ensures that the impact of the unavailable data is removed from the model.
- Apply the computed parameter adjustments to the parameters of the previously trained adapter, effectively updating it to reflect the removal of the unavailable data. This yields the final unlearned model parameters while preserving efficiency and minimizing retraining overhead.

## 4 EXPERIMENT

In this section, we carry out extensive experiments to assess the performance and efficiency of LLMEraser. The experiments are designed to explore the following key research questions: **RQ1:** How does LLMEraser perform across various unlearning tasks? **RQ2:** How does LLMEraser perform at different unlearning ratios? **RQ3:** How does the efficiency of LLMeraser compared to other unlearning methods?

### 4.1 EXPERIMENTAL SETUPS

We conduct experiments on both LLMs and Multimodal Large Language Models (MLLMs), focusing specifically on LLMs for Recommendation (LLM4Rec) (Bao et al., 2023; Liao et al., 2024) and MLLM relation mining tasks (Wu et al., 2024c; Ye et al., 2024), to validate the effectiveness of our proposed LLMEraser. We choose LLaMA2-7B (Touvron et al., 2023b) as our backbone LLM and LLaVA 1.5-7B (Liu et al., 2023a) for the MLLM experiments. Comprehensive details on task, datasets, baselines, and evaluation metrics for our proposed LLMEraser can be found in Appendix C.

### 4.2 RESULTS ANALYSIS FOR VARIOUS UNLEARNING TASKS (RQ1)

We design a variety of comprehensive experiments to thoroughly validate the effectiveness of LLMEraser across the three unlearning tasks we have proposed.

#### 4.2.1 RESULTS ANALYSIS ON INSTANCE REMOVAL

For instance removal, we directly delete a proportion of training instances and subsequently evaluate the performance of each unlearning method. The experimental results on LLM4Rec are shown in Table 2. We can find that: (1) LLMEraser closely mirrors the performance of Retrain. The performance gap between LLMEraser and Retrain is merely 0.0038, constituting only 0.6% of Retrain's performance. This can be attributed to our method's direct estimation of the parameter changes between the retrained model and the original model, allowing for a highly accurate calculation of these changes. (2) Other unlearning methods exhibit notable declines in model performance. Specifically, Gradient Ascent and E2URec show average decreases of 2.7% and 2.4%, respectively, as they do not explicitly aim to approximate the Retrain model during the fine-tuning process.

---

[1]HVP has a corresponding implementation in PyTorch; refer to https://pytorch.org/docs/stable/autograd.html for details.

### 4.2.2 RESULTS ANALYSIS ON QUERY MODIFICATION & RESPONSE CORRECTION

Adversarial attack experiments are widely employed to assess the efficacy of data modification for unlearning techniques (Wu et al., 2023; Moon et al., 2024; Cha et al., 2024). The core idea is first randomly introducing corrupted instances into the dataset, which inevitably leads to a decline in model performance, and then leveraging unlearning techniques to correct these noisy data on the model. Following this setting, we evaluate the performance of LLMEraser in both query modification and response correction tasks.

For query modification, we conduct experiments on the LLM4Rec task by adding adversarial noise to the user interaction sequences, *i.e.,* randomly deleting some items from the sequences (Interaction Removal) or replacing them with corrupt ones (Interaction Replacement), and then using LLMEraser to rectify the data. Table 3 presents the experimental results. We can observe that: (1) LLMEraser brings a substantial utility gain to the model compared to the corrupted baseline, significantly reducing the negative impact of noisy data. Specifically, it achieves an average improvement of 5.1% compared to the corrupted model in both settings, with a peak increase of 5.5% in interaction removal setting. Moreover, its performance is closest to that of Retrain, demonstrating its effectiveness in correcting inaccurate input information. (2) SISA and RecEraser fail to improve performance. Their average results in both settings decreased by 7.0% and 31.3% compared to the corrupted baseline. The reasons may lie in their dataset partitioning and submodel retraining strategy, potentially leading to a loss of crucial contextual information and introducing inconsistencies in learned representations. (3) RecEraser underperforms SISA in most cases. Designed on traditional recommendation models, RecEraser relies on users' collaborative signals to optimize shard partitioning; however, this strategy fails to effectively adapt to LLM4Rec.

For response correction, we introduce noise into the training data of the MLLMs task by randomly assigning incorrect labels to a portion of the samples. In the spurious biases task for MLLMs, we reverse 40% the original "yes/no" labels. For the hard hallucination mining task in MLLMs, we assign random labels to 40% of the samples. We leverage LLM unlearning to mitigate the negative impact of such noisy data, aiming to approximate the performance of retraining with clean data. The experimental results of response correction unlearning task on spurious biases task and hard hallucination mining task are presented in Table 4 and 5, respectively. We can draw the following observations: (1) LLMEraser effectively performs response correction, achieving average improvements of 14.2% and 18.9% on the spurious biases task and hard hallucination mining task, respectively, compared to the corrupted baseline. Compared to other methods, LLMEraser shows the smallest performance gap relative to Retrain. On the spurious biases task and hard hallucination mining task, the average differences with Retrain are 0.024 and 0.048, which account for 2.9% and 7.5% of Retrain's performance, respectively. Whether addressing label reversal in binary classification or correcting labels in multi-class scenarios, LLMEraser can eliminate the negative impact of noisy labels and restore them to their clean, original state. (2) The improvement brought by SISA is not significant. Although SISA ensures that dirty data is replaced with clean data during retraining, its data segmentation strategy can inevitably hurt model performance.

### 4.3 RESULTS ANALYSIS FOR DIFFERENT UNLEARNING RATIOS (RQ2)

To assess the sensitivity of various unlearning methods to different scales of unlearning data, we conduct experiments using different unlearning ratios in instance removal and query modification tasks. For the instance removal, we employ TallRec as the LLM4Rec framework, where 5% and 10% of instances are removed. Meanwhile, for query modification, LLARA is utilized as the backbone, where 5% and 10% of user interactions are deleted. The experimental results are shown in Figure 3. From these results, we can find that: (1) In the instance removal task, LLMEraser consistently performs closest to Retrain across different unlearning ratio settings, with an average performance decline of only 1.18%. This indicates that LLMEraser can effectively delete data while minimizing the neg-

Table 2: Experimental results on the instance removal task with 5% of training data removed, using TALLRec as the LLM4Rec model on the BookCrossing dataset.

|  | Original | Retrain | Gradient Ascent | E2URec | **LLMEraser (Ours)** |
|---|---|---|---|---|---|
| AUC | 0.6400 | 0.6357 | 0.6187 | 0.6205 | **0.6319** |

Table 3: Experimental results on the QM task, using LLaRA as the LLM4Rec model on the Movie-Lens and LastFM datasets. "10% Interaction Removal" refers to 10% of users have items removed from their interaction sequences, "5% Interaction Replacement" refers to 5% of users have items replaced with noisy interactions. **Corrupted** refers to the model trained with the noisy data.

| Method | | Movielens | | LastFM | |
|---|---|---|---|---|---|
| | | HitRatio@1 | ValidRatio | HitRatio@1 | ValidRatio |
| 10% Interaction Removal | Retrain | 0.4565 | 0.9684 | 0.4508 | 1.0000 |
| | Corrupted | 0.4222 | 0.9375 | 0.4344 | 1.0000 |
| | SISA | 0.4130 | 0.9684 | 0.4132 | 0.9918 |
| | RecEraser | 0.2717 | 0.9684 | 0.4298 | 0.9918 |
| | **LLMEraser (Ours)** | **0.4456** | **0.9684** | **0.4463** | 0.9918 |
| 5% Interaction Replacement | Retrain | 0.4565 | 0.9684 | 0.4508 | 1.0000 |
| | Corrupted | 0.4316 | 0.9684 | 0.4344 | 0.9918 |
| | SISA | 0.3804 | 0.9684 | 0.4050 | 0.9918 |
| | RecEraser | 0.3152 | 0.9684 | 0.3689 | 1.0000 |
| | **LLMEraser (Ours)** | **0.4516** | **0.9789** | **0.4426** | **1.0000** |

Table 4: Experimental results on the MM-SPUBENCH for RC tasks, where **Corrupted** denotes we assign wrong labels for 40% of the training samples.

| Method | MM-SPUBENCH | | | | | | | | | Average | All |
|---|---|---|---|---|---|---|---|---|---|---|---|
| | BG | TN | CO | RS | Col. | Ori. | LS | PA | Sha. | | |
| Retrain | 0.88 | 0.80 | 0.83 | 1.00 | 0.78 | 0.86 | 0.86 | 0.66 | 0.70 | 0.82 | 0.84 |
| Corrupted | 0.76 | 0.62 | 0.67 | 0.80 | 0.67 | 0.76 | 0.65 | 0.68 | 0.67 | 0.70 | 0.71 |
| SISA | 0.84 | 0.65 | 0.79 | 1.00 | 0.64 | 0.79 | 0.86 | 0.73 | 0.57 | 0.76 | 0.77 |
| **LLMEraser** | **0.86** | **0.70** | **0.80** | **1.00** | **0.78** | **0.85** | **0.84** | **0.76** | **0.67** | **0.81** | **0.81** |

ative impact on model performance. (2) In the query modification task, LLMEraser consistently achieves the best performance across various unlearning ratios, with an average improvement of 4.9% compared to corrupted method. Notably, at an unlearning ratio of 10%, the relative improvement reaches 5.1%. The average difference between LLMEraser and Retrain is only 0.0079. In comparison to SISA and RecEraser, LLMEraser demonstrates a superior ability to maintain model utility. This highlights the effectiveness of LLMEraser, demonstrating its robust performance across varying unlearning demands. (3) We observe an interesting phenomenon in query modification task under adversarial attack settings, with a sufficiently high unlearning ratio (in this case, 5% and 10%), both SISA and Receraser require retraining all shards with the same clean data, resulting in equivalent outcomes. Despite the direct use of clean data for retraining, they still struggle to obtain optimal model performance.

## 4.4 RESULTS ANALYSIS FOR UNLEARNING EFFICIENCY (RQ3)

Efficiency is a key metric in evaluating unlearning techniques, particularly for LLMs. We here conduct experiments, comparing our proposed LLMEraser against existing techniques. For a fair comparison, we report the execution time in the QM task, where 5% of users have items replaced with noisy interactions. All methods are run on a single Nvidia A100 GPU. Table 6 presents the results. We can observe that: (1) Due to the parallel training of sub-models, the retraining time of both SISA and RecEraser can be reduced to some extent. However, RecEraser requires data partitioning based on similarity, which introduces additional computational overhead. Moreover, both methods remain highly inefficient as unlearning requests necessitate retraining of the adapters. (2) In contrast, our proposed LLMEraser exhibits remarkable efficiency in handling unlearning tasks. By directly modifying model parameters, LLMEraser achieves a speedup of approximately 31.25 times compared to retraining, requiring only about $1.4 \times 10^3$ seconds to update the parameters. This reduction in execution time demonstrates the effectiveness of our approach in accelerating the computation of

Table 6: Execution time in the QM task.

| Method | Time (s) |
|---|---|
| Retrain | $5.4 \times 10^4$ |
| SISA | $1.8 \times 10^4$ |
| RecEraser | $2.0 \times 10^4$ |
| **LLMEraser** | $1.4 \times 10^3$ |

Table 5: Experimental results on the R-BENCH for RC tasks, where **Corrupted** denotes we assign wrong labels for $40\%$ of training samples.

| Method | Recall | F1-Score | Precision | Accuracy | Yes |
|---|---|---|---|---|---|
| Retrain | 0.70 | 0.66 | 0.63 | 0.65 | 0.55 |
| Corrupted | 0.47 | 0.50 | 0.53 | 0.54 | 0.44 |
| SISA | 0.47 | 0.49 | 0.52 | 0.52 | 0.45 |
| **LLMEraser (Ours)** | **0.68** | **0.63** | **0.58** | **0.56** | **0.50** |

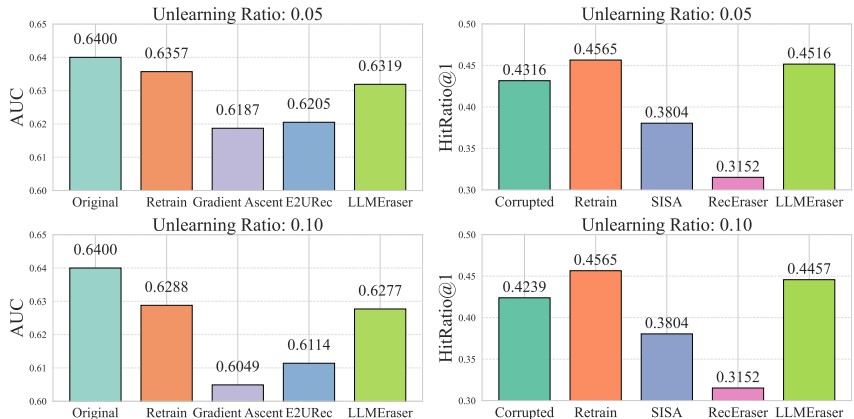

(a) Impact of unlearning ratio in IR.     (b) Impact of unlearning ratio in QM.

Figure 3: 3a: Experimental results of the instance removal task using TallRec as the LLM4Rec model on the BookCrossing dataset, where 5% and 10% of the training data were randomly deleted. 3b: Experimental results of the query modification task using LLaRA as the LLM4Rec model on the MovieLens dataset, where interactions were randomly removed from 5% and 10% of users.

parameter changes. Additional experimental results and related analyses on the memory usage and execution time of LLMEraser can be found in Appendix E.

## 5 LIMITATIONS

LLMEraser offers efficient parameter updates without the need for retraining, making it versatile across different unlearning tasks while also reducing computational overhead. Despite the improvements brought by LLMEraser, its potential shortcomings should not be overlooked. Calculating parameter changes for different unlearning tasks requires accessing the gradient information of the target data and assumes the availability of the training set. Furthermore, the influence function's reliance on the first-order Taylor expansion of the optimization objective leads to inevitable estimation errors, representing an inherent limitation of such an approach.

## 6 CONCLUSION AND FUTURE WORK

This paper introduces LLMEraser, a unified parameter-efficient unlearning framework. By systematically categorizing and addressing various unlearning tasks, LLMEraser leverages influence functions for parameter adjustments, circumventing the cumbersome retraining processes common in traditional methods. Extensive experiments on benchmark datasets show that LLMEraser excels in efficiently handling various unlearning tasks while preserving the overall integrity and efficacy of the models. Additionally, LLMEraser opens new avenues for future research, encouraging the exploration of enhanced unlearning techniques and their implications in diverse applications, such as data privacy and ethical AI. Future studies could explore the broader applicability of LLMEraser and potential optimizations for its computational efficiency and accuracy.

ACKNOWLEDGEMENTS

This research is supported by the National Science and Technology Major Project (2023ZD0121102), National Natural Science Foundation of China (92270114, 62302321, U24B20180, 62121002). The work of Yancheng Yuan is supported by the Research Center for Intelligent Operations Research and The Hong Kong Polytechnic University under grant P0045485. This research is also supported by the advanced computing resources provided by the Supercomputing Center of the USTC.

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

## A  OVERVIEW OF EXISTING LLM UNLEARNING METHODS

- **SISA (Bourtoule et al., 2021):** It works by dividing the training dataset into partitions, allowing for targeted unlearning of specific instances. The methodology typically involves the following steps: data partitioning, retraining, and aggregation. However, a notable limitation of SISA is that it does not preserve the model architecture and requires retraining of sub-models, which can lead to increased computational costs.

- **FairSISA (Kadhe et al., 2023):** FairSISA improves upon SISA by incorporating fairness enhancements. It still relies on the paradigm of retraining sub-models to handle unlearning requests. This approach inherently alters the model architecture and necessitates the retraining of the sub-models, which can limit the flexibility and efficiency of the unlearning process.

- **APA (Hu et al., 2024c):** This study introduces the first exact unlearning approach for large language model-based recommendation (LLMRec), focusing on the removal of personal data to comply with privacy regulations. The Adapter Partition and Aggregation (APA) method is proposed, which combines data partitioning with parameter aggregation to reduce inference latency while maintaining performance. This approach enables efficient unlearning without incurring the extra costs typically associated with traditional methods. However,it can affect the integrity of the adapter structure and necessitates retraining of sub-models.

- **Gradient Ascent:** It utilizes the gradient of the target instance to fine-tune the adapter by moving in the direction of the negative gradient of the deleted data. However, this approach is not effective for input modification and output correction tasks, as gradient ascent of target instances cannot adequately handle these scenarios.

- **EUL (Chen & Yang, 2023):** This work introduces a lightweight approach for LLMs to efficiently forget specific information without complete retraining. It incorporates unlearning layers into transformer architectures, utilizing a selective teacher-student formulation, and employs a fusion mechanism to combine multiple unlearning layers into a unified layer. This enables LLMs to dynamically handle a sequence of deletion requests while maintaining model performance. The introduction of adapters alters the model's structure, and the KL divergence-based methods are only effective for instance removal tasks, as obtaining a model trained on clean data is not feasible.

- **E2URec (Chen & Yang, 2023):** This method uses lightweight LoRA modules and a teacher-student framework to forget specific data while maintaining performance. However, the extra LoRA module changes the original model architecture, and the teacher-student framework requires pretraining on both retained and forgotten data, which is intricate and cannot perform well on other tasks like editing.

## B  ALGORITHM FOR CALCULATING PARAMETER CHANGES

The algorithm for calculating the parameter changes $\Delta\Theta_{\text{Task}}$ can be found in Algorithm 1. This algorithm accelerates the computation of parameter changes resulting from unlearning requests and is applicable in large-scale data scenarios.

---

**Algorithm 1** Calculate Parameter Changes $\Delta\Theta_{\text{Task}}$

---

1: **Input:** target_data, train_data_loader, old_adapter, loss_fun, $n$, Task, $\Delta_{init}$, $\Delta_{lr}$
2: **Output:** Parameter changes $\Delta\Theta_{\text{Task}}$
3: **if** Task = IR **then**
4:     $b \leftarrow \frac{1}{n}\sum_{(x,y)\in\mathcal{S}_{\text{IR}}}\mathcal{G}(x,y)$
5: **else if** Task = RC **then**
6:     $b \leftarrow \frac{1}{n}\sum_{(x,y)\in\mathcal{S}_{\text{IM}}}\mathcal{G}(x,y) - \frac{1}{n}\sum_{(x+\delta_x,y)\in\mathcal{S}_{\text{IM}}}\mathcal{G}(x+\delta_x,y)$
7: **else if** Task = IM **then**
8:     $b \leftarrow \frac{1}{n}\sum_{(x,y)\in\mathcal{S}_{\text{RC}}}\mathcal{G}(x,y) - \frac{1}{n}\sum_{(x,y+\delta_y)\in\mathcal{S}_{\text{RC}}}\mathcal{G}(x,y+\delta_y)$
9: **end if**
10: $\Delta \leftarrow \text{initialize}(\Delta_{init})$
11: $optimizer \leftarrow \text{Adam}([\Delta], lr = \Delta_{lr})$
12: **while** not converge **do**
13:     $data \leftarrow \text{get\_batch}(train\_data\_loader)$
14:     $batch\_loss \leftarrow \text{loss\_fun}(data.x, data.y)$
15:     $batch\_grad \leftarrow \nabla(batch\_loss, old\_adapter.parameters())$
16:     $hvp \leftarrow \nabla(batch\_grad, old\_adapter.parameters(), \text{output} = b)$
17:     $optimizer.zero\_grad()$
18:     $funv\_value \leftarrow \frac{1}{2}\cdot\langle hvp, p\rangle - \langle b, p\rangle$
19:     $funv\_value.backward()$
20:     $optimizer.step()$
21: **end while**
22: **Return** Parameter changes $\Delta\Theta_{\text{Task}} = \Delta$

---

## C  EXPERIMENTAL DETAILS

In this section, we briefly introduce the tasks used to validate LLMEraser on the unlearning tasks for IR, QM, and RC, as discussed in Section 4. These tasks are designed to assess LLMEraser's effectiveness in handling unlearning scenarios, where specific instances or data are removed or corrected when certain unlearning request arrives.

- For LLM4Rec unlearning tasks, our implementation is based on two representative PEFT methods: TallRec (Bao et al., 2023) for item rating, and LLaRA (Liao et al., 2024) for item ranking. Specifically, we frame the rating tasks (TallRec) as a binary classification problem, predicting whether or not the user prefers a target item. We employ AUC as the evaluation metric. For the ranking tasks (LLaRA), which recommend items to users from a candidate set, we utilize HitRatio@1 and ValidRatio to evaluate the relevance of recommended items among all candidates and the proportion of effective responses separately.

- In terms of MLLMs unlearning tasks, we focus on hard hallucination mining, *e.g.,* understanding of relation (Wu et al., 2024c) and spurious biases (Ye et al., 2024). We structure the evaluation as binary or multi-choice classification problems, which aim to select the ground-truth from the noisy labels. Specifically, for relation understanding, we follow (Wu et al., 2024c) to present the Recall, F1-Score, Precision, Classification accuracy, Yes ratio as the evaluation metrics. For spurious biases, we follow (Wu et al., 2024c) to show the classification accuracy for 9 types of spurious correlations, which is Background (BG), Texture and Noise (TN), Co-occurring Objects (CO), Relative Size (RS), Colorization (Col.), Orientation (Ori.), Lighting and Shadows (LS), Perspective and Angle (PA), and Shape (Sha.).

**Datasets:** Our experimental datasets for LLM4Rec unlearning tasks include three commonly used recommendation datasets: BookCrossing (Ziegler et al., 2005), MovieLens (Harper & Konstan,

2016), and LastFM (Cantador et al., 2011). We follow the data preprocessing and dataset partitioning as described in (Bao et al., 2023) and (Liao et al., 2024). For MLLMs unlearning tasks, we utilize MMSpuBench (Ye et al., 2024), and R-Bench (Wu et al., 2024c) with the representative masked instances for evaluation, partitioning the data is into training (60%), validation (20%), and testing (20%) set.

**Baselines:** We carefully select the following methods for comparison. **Original**: The original model without unlearning modifications. **Retrain**: It retrains the adapters using the dataset after correction or removal. **SISA** (Sekhari et al., 2021): It divides the training data into disjoint shards and subsequently retrains sub-models (adapters) associated with the shards containing unlearning data. **RecEraser** (Chen et al., 2022): An enhancement of SISA, refining the aggregation strategy and taking into account collaborative signals during data partitioning. **Gradient Ascent**: It finetunes adapters using the reverse gradients of the deleted data. **E2URec** (Wang et al., 2024): An approach to implement instance removal based on KL divergence within a teacher-student framework.

# D    ESTIMATION ERRORS ANALYSIS OF LLMERASER

The approximation errors in LLMEraser consist of two primary components: first, the errors introduced by the Taylor expansion approximation in the derivation of the influence function, where high-order terms are neglected; and second, the errors arising from the new algorithm proposed in Section 3.3 for solving the inverse Hessian-vector-product. We will conduct the error analysis in two parts accordingly.

## D.1    ERRORS ANALYSIS FOR TAYLOR EXPANSION APPROXIMATION

Without loss of generality, we consider approximation error in Equation 6. In other words, we will analyze the error $\|\Delta\Theta(\epsilon) + \epsilon H_{\hat{\Theta}}^{-1}\nabla_{\Theta}\mathcal{L}((x, y); \hat{\Theta})\|$.

The derivation below follows from (Zhang et al., 2022), where we assume that $H_{\hat{\Theta}}$ is invertiable. As we discussed in our paper, this can be guaranteed if the second-order sufficient condition holds at $\hat{\Theta}$.

Since $\hat{\Theta}_{\text{new}}(\epsilon)$ is an optimal solution to the perturbed loss function defined in Equation 5, we have

$$\nabla_{\Theta}R(\mathcal{Z}; \hat{\Theta}_{\text{new}}(\epsilon)) + \epsilon\nabla_{\Theta}\mathcal{L}((x, y); \hat{\Theta}_{\text{new}}(\epsilon)) = 0.$$

Since $\hat{\Theta}_{\text{new}}(\epsilon) \approx \hat{\Theta}$ when $\epsilon$ is sufficiently small, it follows from the Taylor expansion that

$$0 = [\nabla_{\Theta}R(\mathcal{Z}; \hat{\Theta}) + \epsilon\nabla_{\Theta}\mathcal{L}((x, y); \hat{\Theta})] + [H_{\hat{\Theta}} + \epsilon\nabla_{\Theta}^2\mathcal{L}((x, y); \hat{\Theta})]\Delta\Theta(\epsilon) + o(\|\Delta\Theta(\epsilon)\|).$$

Since $\hat{\Theta}$ is an optimal solution to the loss function defined in Equation 3, we have $\nabla_{\Theta}R(\mathcal{Z}; \hat{\Theta}) = 0$. Therefore,

$$\Delta\Theta(\epsilon) = -[H_{\hat{\Theta}} + \epsilon\nabla_{\Theta}^2\mathcal{L}((x, y); \hat{\Theta})]^{-1}(\epsilon\nabla_{\Theta}\mathcal{L}((x, y); \hat{\Theta}) + o(\|\Delta\Theta(\epsilon)\|).$$

Since $\hat{\Theta}$ is an optimal solution to the loss function defined in Equation 3, $H_{\hat{\Theta}}$ is positive semidefinite. Therefore, the assumption that $H_{\hat{\Theta}}$ is invertiable implies that $H_{\hat{\Theta}}$ is positive definte. Therefore, we know that

$$\Delta\Theta(\epsilon) = -H_{\hat{\Theta}}^{-1}(\epsilon\nabla_{\Theta}\mathcal{L}((x, y); \hat{\Theta}) + o(|\epsilon|)\|\Delta\Theta(\epsilon)\| + o(\|\Delta\Theta(\epsilon)\|).$$

Therefore, as $\epsilon \to 0$,

$$\|\Delta\Theta(\epsilon) + H_{\hat{\Theta}}^{-1}(\epsilon\nabla_{\Theta}\mathcal{L}((x, y); \hat{\Theta})\| = o(|\epsilon|) + o(\|\Delta\Theta(\epsilon)\|) \to 0.$$

In our applications, we know that $\epsilon = O(1/n)$, where $n$ is the number of training samples. Therefore, $\epsilon$ should be very small and our approximation to $\Delta\Theta(\epsilon)$ by the influence function should be accurate for applications with a very large training datasets.

Table 7: Memory usage (measured in megabytes, MB) for different LoRA ranks (8, 16, 32) on the QM task, using LLaRA as the LLM4Rec model on the LastFM dataset, where 10% of users have items replaced with noisy interactions.

| Method | LoRA r = 8 | LoRA r = 16 | LoRA r = 32 |
|---|---|---|---|
| Retrain | 33040 MB | 33868 MB | 34128 MB |
| SISA | 33040 MB | 33868 MB | 34128 MB |
| **LLMEraser (Ours)** | **30760 MB** | **31386 MB** | **31834 MB** |

## D.2    ERRORS ANALYSIS FOR OUR PROPOSED ALGORITHM

For our proposed Algorithm, the estimation errors analysis is as follows. For a given (approximate) solution $\widetilde{\Delta}$ to the Equation 12, the error is defined as

$$err(\widetilde{\Delta}) := \|\nabla^2_{\widehat{\Theta}} R(\mathcal{Z}; \widehat{\Theta})\widetilde{\Delta} - b\| = \|\nabla F(\widetilde{\Delta})\|,$$

where the function $F(\cdot)$ is defined in Equation 14. Therefore, the theoretical analysis of $err(\widetilde{\Delta})$ is equivalent to the error analysis of $\|\nabla F(\Delta_t)\|$ for the sequence $\{\Delta_t\}_{t\geq 1}$ generated by the optimization algorithm for solving the problem Equation 9, Equation 10, and Equation 11.

Since we use ADAM as a default optimizer for solving Equation 9, Equation 10, and Equation 11, we analyze the error $\|\nabla F(\Delta_t)\|$ for the sequence $\{\Delta_t\}_{t\geq 1}$ generated by ADAM. It follows from (Zhang et al., 2022) that ADAM can converge without modifications if the hyper-parameters are appropriately chosen (say the default choice $\beta_1 = 0.9$, $\beta_2 = 0.999$).

Moreover, under reasonable assumptions (see (Zhang et al., 2022) for more details), it holds that

$$\min_{k_m \leq t \leq T} \mathbb{E}\|\nabla F(\Delta_t)\|_2 = \mathcal{O}(\log T/\sqrt{T}) = \widetilde{\mathcal{O}}(1/\sqrt{T}).$$

Since for sufficiently large $T$, $\log T < T^q$ for any $q > 0$, we know we can achieve

$$\min_{k_m \leq t \leq T} \mathbb{E}\|\nabla F(\Delta_t)\|_2 \leq \epsilon$$

for small $\epsilon > 0$ in $\widetilde{\mathcal{O}}(\epsilon^{-2}) \approx O(\epsilon^{-2})$ iterations. This proof also ensures the convergence of the algorithm proposed in Section 3.3.

## E    DISCUSSION ABOUT THE EFFICIENCY OF LLMERASER

Our proposed algorithm in Section 3.3 for computing the parameter changes not only accelerates the calculation of parameter changes but also significantly reduces GPU memory consumption. As highlighted in our paper, while Conjugate Gradients (CG) is an effective method for computing parameter changes, it requires full-batch computation (Agarwal et al., 2016), which is infeasible for LLMs. Our new algorithm overcomes this limitation, making it practical to compute adapter's parameter changes in the context of LLMs.

Specifically, LLMEraser formulates the parameter updates as an inverse Hessian-vector-product (Equation 9, Equation 10, and Equation 11). Importantly, although the inverse Hessian appears in the formulation, it does not require explicit computation or inversion of the Hessian matrix. Directly calculating the inverse Hessian-vector-product has a time complexity of $O(p^3)$ and a space complexity of $O(p^2)$, as the Hessian matrix needs to be stored—making it highly memory-intensive.

Our method transforms the computation of the inverse Hessian-vector-product into the problem of solving for the Hessian-vector-product, enabling efficient resolution through mini-batch algorithms. The Hessian-vector-product, if computed directly via the full Hessian matrix multiplication, would have a time and space complexity of $O(p^2)$. However, using HVP (Hessian-free methods), we avoid the explicit computation and storage of the Hessian matrix, reducing both time and space complexity to $O(p)$ (Pearlmutter, 1994). By further leveraging mini-batch optimization for Equation 12, LLMEraser achieves a space complexity of $O(p)$, ensuring its scalability.

The results for the LastFM dataset using the LLaRA backbone with LoRA ranks of 8, 16, and 32 are shown in the Table 8.

Table 8: Experimental results on the QM task for different LoRA ranks (8, 16, 32), using LLaRA as the LLM4Rec model on the LastFM dataset, where 10% of users have items replaced with noisy interactions. "Corrupted" refers to the model trained with the noisy data.

| Method | LoRA r = 8 | | LoRA r = 16 | | LoRA r = 32 | |
| --- | --- | --- | --- | --- | --- | --- |
| | HitRatio@1 | ValidRatio | HitRatio@1 | ValidRatio | HitRatio@1 | ValidRatio |
| Retrain | 0.4508 | 1.0000 | 0.4417 | 0.9836 | 0.4215 | 0.9918 |
| Corrupted | 0.4344 | 0.9918 | 0.4098 | 1.0000 | 0.4016 | 1.0000 |
| **LLMEraser** | **0.4426** | **1.0000** | **0.4344** | **1.0000** | **0.4180** | **1.0000** |

Table 9: Execution time (measured in seconds) for different LoRA ranks (8, 16, 32) on the QM task, using LLaRA as the LLM4Rec model on the LastFM dataset, where 10% of users have items replaced with noisy interactions.

| Method | LoRA r = 8 | LoRA r = 16 | LoRA r = 32 |
| --- | --- | --- | --- |
| Retrain | $1.68 \times 10^4$ | $1.69 \times 10^4$ | $1.69 \times 10^4$ |
| **LLMEraser (Ours)** | $1.50 \times 10^3$ | $1.53 \times 10^3$ | $1.56 \times 10^3$ |

We can observe that LLMEraser effectively reduces the negative impact of noisy data and brings a significant utility gain. The HitRatio@1 improves by an average of 4.9%, and the performance is comparable to that of Retrain. This demonstrates that LLMEraser can effectively forget and correct the adverse effects caused by noisy data.

Regarding GPU memory usage, we measure the GPU utilization of the LLaRA backbone with LoRA rank sets to 8, 16, and 32. The statistical information and the experimental results (with memory usage measured in megabytes (MB)) are shown in Table 7.

The GPU utilization of SISA is identical to that of Retrain because SISA (Kwak et al., 2017) effectively requires retraining all parameters (We report the memory usage required to train a single shard). Similarly, fine-tuning-based methods such as gradient descent also necessitates updating all parameters. The backbone of the LLM we used is LLaMA2-7B (Touvron et al., 2023b).

The runtime results for LoRA with ranks 8, 16, and 32 on the LastFM dataset are shown in Table 9. The evaluation is measured in seconds.

In summary, the time and space complexity of LLMEraser are both $O(p)$, where $p$ represents the number of parameters. This indicates that LLMEraser is highly efficient in terms of both time and space, as its performance scales linearly with the number of parameters. This efficiency makes LLMEraser a suitable choice for real-world applications where computational resources and time are critical considerations.

## F    RELATED WORK

### F.1    LARGE LANGUAGE MODELS

Recent advancements in natural language processing (NLP) (Nam et al., 2024; Jin et al., 2024) have been significantly driven by the development of pretrained language models and Large Language Models. The introduction of models like BERT (Devlin et al., 2019) and GPT-2 (Radford et al., 2019) marked a pivotal shift in leveraging large-scale unsupervised pretraining, enabling superior performance across various NLP tasks through fine-tuning. The scaling of language models led to the emergence of LLMs such as GPT-3 (Brown et al., 2020) and PaLM (Chowdhery et al., 2023), which have pushed the boundaries of language understanding and generation. These models, with billions of parameters, are capable of performing complex reasoning and handling diverse tasks with minimal instruction.

Recent research has explored parameter-efficient fine-tuning techniques, which adapt large models to specific applications without requiring extensive computational resources. Techniques like Adapter modules (Houlsby et al., 2019) and Low-Rank Adaptation (LoRA) (Hu et al., 2022) have gained popularity for their efficiency and effectiveness in maintaining performance while reducing

the number of trainable parameters. Furthermore, instruction tuning (Liu et al., 2023a; Tang et al., 2024) using domain-specific data has emerged as a key strategy to enhance model performance in specialized contexts. Works by Ouyang et al. (2022) and Dodge et al. (2020) illustrate how tailoring models to specific tasks through targeted instruction can significantly improve their utility, particularly in complex domains, demonstrating the importance of context and relevance in model training.

LLMs have found extensive applications in various downstream tasks (Fang et al., 2024b; Hu et al., 2024a; Wu et al., 2024b), demonstrating their versatility across domains such as natural language processing, information retrieval, and knowledge graph augmentation (Zhang et al., 2024a; Xu et al., 2024b; Fang et al., 2024a; Sheng et al., 2024). For instance, LLMs are employed to enhance the accuracy of query-based systems by leveraging their ability to understand and generate contextually relevant responses, improving user experience in search applications (Liu et al., 2024b; Shang & Huang, 2024). Additionally, they are utilized in graph analytics, enabling complex reasoning tasks and facilitating the extraction of insights from structured data (Chen et al., 2023; Xu et al., 2024a). The adaptability of LLMs through prompt engineering further supports their deployment in specific use cases, allowing for tailored outputs that meet diverse requirements (Arawjo et al., 2024; Cain, 2024).

In a similar vein, LLMs are increasingly being integrated into recommendation systems, building on their capabilities in natural language processing and understanding user preferences. Traditional recommendation systems often rely on collaborative filtering (Misztal-Radecka & Indurkhya, 2020; Wu et al., 2024a), content-based approaches (Pazzani & Billsus, 2007; Wu et al., 2022), or hybrid models (Burke, 2002). Recent advances, including Reinforced Prompt Personalization (Mao et al., 2024; Xin et al., 2022), and the incorporation of LLMs into recommendation systems via tool learning (Zhao et al., 2024; Dehbozorgi et al., 2024) or fine-tuning with recommendation-specific data (Kong et al., 2024; Chen et al., 2024), have significantly improved personalization. These methods enable LLMs to better capture user preferences and context (Lyu et al., 2024; Hu et al., 2024b), ultimately enhancing the accuracy and relevance of recommendations.

## F.2 LARGE LANGUAGE MODELS UNLEARNING

The concept of unlearning in Large Language Models has garnered considerable attention as concerns over data privacy and model integrity have intensified. In-context unlearning, proposed by Pawelczyk et al. (2023), allows the selective removal of data points by supplying flipped labels during inference, effectively maintaining performance while unlearning specific information. Additionally, Quark by Lu et al. (2022) employs a reinforcement learning framework to control and reduce undesirable behaviors, enhancing text generation without extensive retraining.

Chen & Yang (2023) introduce a lightweight unlearning method that integrates unlearning layers into transformer architectures, facilitating efficient data removal. Knowledge Unlearning by Jang et al. (2023) demonstrates that targeted gradient ascent can effectively forget sensitive information, surpassing traditional methods in performance retention. The technique proposed by Eldan & Russinovich (2023) facilitates the removal of specific facts related to the Harry Potter series while preserving the model's overall performance.

Other approaches, such as the Partitioned Gradient Update (PGU) method by Yu et al. (2023), aim to reduce social biases effectively. Collectively, these studies underline the significance of unlearning in LLMs, paving the way for safer, more responsible AI applications.

## G MORE EXAMPLES OF VARIOUS UNLEARNING TASKS

**Instance Removal Case Study**

Query
Given the user's historical interactions, please determine whether the user will enjoy the target new movie by answering "Yes" or "No".
User's liked items: GodFather.
User's disliked items: Star Wars.
Target new movie: Iron Man

Response
No.

---

**Query Modification Case Study**

Query
This user has watched: The Rich Man's Wife [emb], Air Force One [emb], Murder at 1600 [emb], Absolute Power in the previous [emb]. Please predict the next movie this user will watch. Choose the answer from the following 10 movie titles: Face/Off [emb], Primal Fear [emb], Ransom [emb], Men in Black [emb], Twelve Monkeys [emb], Lone Star [emb], Mr. Holland's Opus [emb], Jackie Chan's First Strike [emb], Waiting for Guffman [emb], The Long Kiss Goodnight [emb]. Answer:

Response
Face/Off

**After Query Modification**

Query
This user has watched: The Rich Man's Wife [emb], Air Force One [emb], Murder at 1600 [emb], Absolute Power in the previous [emb]. Please predict the next movie this user will watch. Choose the answer from the following 10 movie titles: Face/Off [emb], Primal Fear [emb], Ransom [emb], Men in Black [emb], Twelve Monkeys [emb], Lone Star [emb], Mr. Holland's Opus [emb], Jackie Chan's First Strike [emb], Waiting for Guffman [emb], The Long Kiss Goodnight [emb]. Answer:

Figure 4: Instance Removal Case Study & Query Modification Case Study.

**Response Correction Case Study**

Query    Is the elephant in red mask standing next to a tree in green mask?

Response   Yes

**After Response Correction**

Response   No

---

**Response Correction Case Study**

Query    You are a helpful assistant that can answer questions for an image. I will provide you 4 options.\nResponse Format\nChoice: A single character from A, B, C, D.\nWhich feature best indicates the identity of the object that has a floral pattern and is placed on a chair?\nChoices:A. The object's soft texture\nB. The indoor setting\nC. The wooden chair\nD. The background clutter

Response   D

**After Response Correction**

Response   A

Figure 5: Response Correction Case Study.

