# OpenReview forum: "Unified Parameter-Efficient Unlearning for LLMs"
_ICLR.cc/2025/Conference — ICLR 2025 Poster_

### Official Review · Reviewer_nWJj · 2024-11-02

**Soundness:** 3
**Presentation:** 3
**Contribution:** 3
**Rating:** 6
**Confidence:** 3

**Summary:**

This paper introduces LLMEraser, a unified framework for parameter-efficient unlearning in large language models (LLMs), specifically tailored to address privacy and security concerns in domain-specific fine-tuning. The framework utilizes influence functions to perform instance-wise unlearning tasks such as instance removal, query modification, and response correction. LLMEraser allows unlearning without requiring full model retraining, and has demonstrated efficacy in preserving model performance across various unlearning tasks.

**Strengths:**

1. The paper is written clearly and is easy to follow, with clearly presented formulations.
2. The taxonomy of unlearning tasks is well-defined, providing clarity on the types of unlearning scenarios addressed.
3. Extensive experiments are conducted across diverse unlearning tasks with both LLMs and MLLMs, covering recommendation and multimodal relation mining applications.
4. LLMEraser accelerates the computation of the inverse Hessian-vector-product in the influence function, enabling efficient implementations. This improvement is valuable for LLM applications, where computational efficiency is a growing concern.

**Weaknesses:**

1. While LLMEraser reduces the need for retraining, the memory-intensive computation of inverse Hessian-vector products remains demanding. This requirement may limit scalability, particularly for very large models or environments with limited GPU memory.
2. The method’s reliance on first-order Taylor expansion in influence functions can introduce estimation errors. A lack of detailed error analysis makes it difficult to assess the impact of these errors, especially in tasks requiring high unlearning precision.
3. Limited generalizability is a concern, as the LLM experiments primarily focus on LLMs for recommendation (LLM4Rec). Broader evaluation across more diverse LLM unlearning tasks would strengthen the claims.
4. Key experimental details, such as PEFT configurations and the number of trainable parameters, are not fully provided. These details are essential for evaluating the method's complexity and efficacy across different settings.

**Questions:**

1. Could the authors provide a detailed analysis of estimation errors to demonstrate their impact on performance? Given the importance of precision in unlearning tasks, such an analysis would help establish whether the errors introduced by the method are indeed within an acceptable range and do not significantly detract from the efficacy of LLMEraser in high-precision unlearning scenarios.

2. The LLM4Rec unlearning tasks are different from those unlearning tasks used in the baseline paper. Could the authors provide performance results on additional unlearning tasks more closely aligned with those proposed in other baselines? Additionally, how do the LLM4Rec tasks reflect the generalizability of LLMEraser across diverse unlearning scenarios?

3. While LoRA was used in the experiments, it would be helpful to know the specific experimental setup, including the number of trainable parameters. Since computational complexity is directly related to the number of trainable parameters and rank values in LoRA, could the authors clarify these settings and present results across different ranks? Additionally, could they specify the sample sizes used in unlearning experiments, as sample size likely affects the overall complexity?

4. While the method demonstrates computational efficiency, I am concerned about memory efficiency, particularly for broader applications. Could the authors provide an analysis of the memory cost and specify whether LLMEraser could run on a lower-memory GPU than an Nvidia A100? If the method is limited to high-memory GPUs, this could hinder its practical application. A comparison of memory usage with baseline methods is also important to clarify this point.

I would be happy to engage with the authors to help improve the presentation of the method and evaluation, but my concerns are not insignificant. Clarifications would need to resolve my questions in order for my score to improve.

---

> ### Author Response · Authors · 2024-11-24
> **Response to reviewer nWJj --- Part 1**
>
> We thank the reviewer for the insightful comments. Please find our responses to each point below.
>
> --------
>
> > ### Comment 1 Discussion about memory efficiency
>
> Thanks for your asking! Our proposed algorithm for computing the influence function not only **accelerates** the calculation of parameter changes but also significantly **reduces GPU memory consumption**. As highlighted in our paper, while Conjugate Gradients (CG) is an effective method for computing parameter changes, it requires full-batch computation [3], which is infeasible for LLMs. Our new algorithm overcomes this limitation, making it practical to compute adapter parameter changes using influence functions in the context of LLMs.
>
> Specifically, LLMEraser formulates the parameter updates as an **inverse Hessian-vector product** (equations 9, 10, and 11 in our manuscript). Importantly, although the inverse Hessian appears in the formulation, **it does not require explicit computation or inversion of the Hessian matrix.** Directly calculating the inverse Hessian-vector product has a time complexity of $ O(p^3) $ and a space complexity of $ O(p^2) $, as the Hessian matrix needs to be stored—making it highly memory-intensive.
>
> Our method transforms the computation of the **inverse Hessian-vector product** into the problem of solving for the **Hessian-vector product**, enabling efficient resolution through **mini-batch** algorithms. The Hessian-vector product, if computed directly via the full Hessian matrix multiplication, would have a time and space complexity of $O(p^2)$. However, using **HVP** (Hessian-free methods), we **avoid the explicit computation and storage of the Hessian matrix**, reducing both time and space complexity to **$O(p)$** [1]. By further leveraging mini-batch optimization for equation 12 in the manuscript, LLMEraser achieves a space complexity of **$O(p)$**, ensuring its scalability.
>
> Regarding GPU memory usage, we have measured the GPU utilization of the LLaRA backbone with LoRA rank r sets to 8, 16, and 32. Here are the statistical information and the experimental results (with memory usage measured in megabytes (MB)).
>
> |LoRA r|                                Target modules                               |  #Params  | #Trainable params | Percentage of trainable params |
> |:-----:|:---------------------------------------------------------------------------:|:---------:|:-----------------:|:-------------------------------:|
> |  8 | 'k proj', 'v proj', 'q proj', 'o proj', 'gate proj', 'up proj', 'down proj' | $ 6.7\times10^{10}$  |      $1.9\times10^{8}$     |           0.30%          |
> |16 | 'k proj', 'v proj', 'q proj', 'o proj', 'gate proj', 'up proj', 'down proj' | $ 6.7\times10^{10}$  |      $ 3.9\times10^{8}$      |           0.59%           |
> | 32 | 'k proj', 'v proj', 'q proj', 'o proj', 'gate proj', 'up proj', 'down proj' | $ 6.8\times10^{10}$  |      $ 7.9\times10^{8}$      |           1.17%           |
>
> |     Method    |     LoRA r = 8     |     LoRA r = 16    |     LoRA r = 32    |
> |:-------------:|:---------:|:---------:|:---------:|
> |    Retrain    |   33040 MB |   33868 MB  |   34128  MB |
> |      SISA     |   33040 MB  |   33868 MB  |   34128  MB |
> | **LLMEraser** | **30760 MB** | **31386 MB** | **31834 MB** |
>
>
> The GPU utilization of SISA is identical to that of Retrain because SISA effectively requires retraining all parameters (We report the memory usage required to train a single shard). Similarly, fine-tuning-based methods such as gradient descent also necessitates updating all parameters. The backbone of the LLM we used is LLaMA 2-7B. Our method is highly memory-efficient and can be executed on a single **A40** GPU.
>
> Please let us know if this addresses your question. We are happy to provide additional explanation if required.
>
>
> --------

---

> ### Author Response · Authors · 2024-11-24
> **Response to reviewer nWJj --- Part 2**
>
> >### Comment 2 Specific experimental setup
>
> Thanks for your suggestion! We will provide statistics for the LoRA hyperparameters, details about the datasets, and experimental results for different LoRA ranks. Additionally, we will discuss the scalability of LLMEraser.
>
> - The LoRA configurations and hyperparameters for each task are as follows:
>
> | Method/Dataset | LoRA r | LoRA alpha | LoRA dropout |                                Target modules                               |  #Params  | #Trainable params | Percentage of trainable params |
> |:--------------:|:------:|:----------:|:------------:|:---------------------------------------------------------------------------:|:---------:|:-----------------:|:------------------------------:|
> |     TALLRec    |    8   |     16     |     0.05     |                              'q proj','v proj'                              | $6.7\times10^{10}$ |      $4.2\times10^{7}$     |              0.06%             |
> |      LLaRA     |    8   |     32     |      0.10     | 'k proj', 'v proj', 'q proj', 'o proj', 'gate proj', 'up proj', 'down proj' | $6.7\times10^{10}$ |      $1.9\times10^{8}$     |              0.30%             |
> |   MM-SPUBENCH  |    8   |     16     |     0.05     | 'k proj', 'v proj', 'q proj', 'o proj', 'gate proj', 'up proj', 'down proj' | $7.1\times10^{10}$ |      $4.1\times10^{8}$     |              0.58%             |
> |     R-BENCH    |    8   |     16     |     0.05     | 'k proj', 'v proj', 'q proj', 'o proj', 'gate proj', 'up proj', 'down proj' | $7.1\times10^{10}$ |      $4.1\times10^{8}$     |              0.58%             |
>
> - The results for the LastFM dataset using the LLaRA backbone with LoRA ranks of 8, 16, and 32 are shown in the table below:
>
> |                     | LoRA r = 8 |            | LoRA r = 16 |            |  LoRA r = 32 |            |
> |:-------------------:|:----------:|:----------:|:-----------:|:----------:|:----------:|:----------:|
> |        Method       | HitRatio@1 | ValidRatio |  HitRatio@1 | ValidRatio | HitRatio@1 | ValidRatio |
> |       Retrain       |   0.4508   |   1.0000   |    0.4417   |   0.9836   |   0.4215   |   0.9918   |
> |      Corrupted      |   0.4344   |   0.9918   |    0.4098   |   1.0000   |   0.4016   |   1.0000   |
> | **LMEraser (Ours)** | **0.4426** | **1.0000** |  **0.4344** | **1.0000** | **0.4180** | **1.0000** |
>
>
> We can observe that LLMEraser effectively reduces the negative impact of noisy data and brings a significant utility gain. The HitRatio@1 improves by an average of **4.9%**, and the performance is comparable to that of Retrain. This demonstrates that LLMEraser can effectively forget and correct the adverse effects caused by noisy data.
>
>
> The runtime results for LoRA with ranks 8, 16, and 32 on the LastFM dataset are shown in the table below. The evaluation is measured in seconds (s).
>
> |  Method  | LoRA r = 8         | LoRA r = 16        | LoRA r = 32        |
> |-----------|--------------------|--------------------|--------------------|
> | Retrain   | $ 1.68 \times 10^{4} $   |  $  1.69 \times 10^{4} $ |  $ 1.69 \times 10^{4}  $ |
> | LLMEraser |  $ 1.50 \times 10^{3} $  |  $ 1.53 \times 10^{3} $  |  $ 1.56 \times 10^{3} $  |
>
> - The information regarding the dataset size is as follows:
>
> |   Dataset  | # samples |
> |:------------:|:-----:|
> | BookCrossing | 19414 |
> |   MovieLens  | 47872 |
> |    LastFM    | 66126 |
> |  MM-SpuBench |  2400 |
> |    R-Bench   |  1742 |
>
> Regarding the experimental setup and dataset details, we will provide additional information in the appendix of the revised version. We would like to highlight that LLMEraser introduces **a new algorithm** for computing the influence function, which allows results in the form of the inverse Hessian-vector product to be rewritten as the solution to the problem (12) in our manuscript. Due to its **summative nature**, this can be efficiently solved using **mini-batch algorithms** such as SGD, making our method **applicable to large-scale datasets**. We will also provide the proof of convergence for the algorithm in Comment 4, and this will be included in the appendix of the revised version.
>
> --------

---

> ### Author Response · Authors · 2024-11-24
> **Response to reviewer nWJj --- Part 3**
>
> >### Comment 3 Performance on generative tasks
>
> Thanks for asking! In fact, LLM4Rec can also be regarded as a generative task. For LLaRA, it requires generating the title of the next item the user may interact with. We also agree that generative tasks are highly significant for LLMs. To further demonstrate the generalization ability of LLMEraser, we conducted additional experiments on other tasks, including dialogue summarization and question-answering tasks.
>
> ### **1. Results of the Instance Removal (IR) task**
>
> For **Instance Removal (IR) task**, we conduct experiments on Alpaca-LoRA for the instruction tuning task. The backbone of LLM is LLaMA 2-7B. We remove 5% of the training samples and evaluate the performance of the original model, retrained model, and LLMEraser on the deleted data using ROUGE scores as the evaluation metric. Here are the experimental results.
>
> |    Method    |  ROUGE-1  |  ROUGE-2  |  ROUGE-L  |
> |:-------------:|:---------:|:---------:|:---------:|
> |    Original   |   44.23   |   27.09   |   34.88   |
> |    Retrain    |   39.84   |   20.40   |   31.78   |
> | **LLMEraser** | **39.55** | **20.17** | **31.63** |
>
>
> We can observe that LLMEraser closely matches the performance of Retrain. This is attributed to its direct estimation of parameter changes between the retrained model and the original model, enabling accurate calculations of these changes.
>
> ### **2. Results of the Query Modification (QM) and Response Correction (RC) tasks**
>
> For **Query Modification (QM) and Response Correction (RC) tasks**, we have added experimental results on the DIALOGSUM [4] dataset for dialogue summarization. The backbone of the LLM is FLAN-T5 Base due to time constraints. We will include additional experimental results in the future. Specifically, we apply perturbations to 50% of the samples in the dialogue and 20% of the samples in the summary, and use LLMEraser to correct the corrupted data. Here are the experimental results.
>
> |                         |      Method         | ROUGE-1  | ROUGE-2   | ROUGE-L  |
> |-------------------------|----------------------|-----------|-----------|-----------|
> |                         | Retrain            | 36.59   | 12.78   | 29.56   |
> | 50% Dialogue Distortion | Corrupted            | 34.09     | 11.61     | 26.66     |
> |                         | **LLMEraser (Ours)** | **35.92** | **12.22** | **28.98** |
> |                         | Retrain            | 36.59   | 12.78   | 29.56   |
> | 20% Summary Distortion  | Corrupted            | 35.72     | 11.67     | 29.09     |
> |                         | **LLMEraser (Ours)** | **36.34** | **12.49** | **29.45** |
>
> For MLLM, we conduct experiment on the mm-vet-v2 dataset for integrated capability evaluation task [5]. The data format of mm-vet-v2 is actually in the form of text-based question-answer pairs. The backbone of MLLM is LLaVA 1.5-7B. Specifically, we randomly select 80% of mm-vet-v2 samples as training set, and employ the left 20% samples for evaluation. We apply perturbations to 50% of the training samples and evaluated the performance of the retrained model, corrupted model and LLMEraser on the testing set, where LLMEraser corrects the corrupted data. Here are the experimental results on the **Query Modification (QM)** task, where we utilize "rec, gen, ocr, spat, know" capacities for evaluation, and report the average results. (All Experimental scores are calculated with gpt-4-turbo by following [5].)
>
>
> |                      |    rec   |    gen   |    ocr   |   spat  |   know  |   Average |
> |:--------------------:|:--------:|:--------:|:--------:|:--------:|:--------:|:--------:|
> |        Retrain       |  37.8 |  28.9  |  28.3 |  37.7  |  21.6  |    30.9 |
> |       Corrupted      |   29.4   |   23.0   |   20.7   |   34.1   |   14.1   |   24.3   |
> | **LLMEraser (Ours)** | **33.1** | **27.0** | **21.5** | **37.2** | **17.5**| **27.3** |
>
> LLMEraser provides a substantial utility improvement to the model compared to the corrupted baseline, effectively mitigating the negative impact of noisy data. On the LLM dialogue summarization task, LLMEraser achieves an average improvement of **6.45%** on QM tasks and **3.34%** on RC tasks compared to the corrupted baseline. For MLLM QA tasks, LLMEraser achieves an average improvement of **12.35%** on QM task. Furthermore, LLMEraser’s performance is close to that of Retrain, highlighting its effectiveness in correcting inaccurate information.
>
> In the revised version, we will include these experimental results and implementation details.
>
> --------

---

> ### Author Response · Authors · 2024-11-24
> **Response to reviewer nWJj --- Part 4**
>
> >### Comment 4 Estimation errors analysis
>
> Thanks for your comment!
>
> The approximation errors in LLMEraser consist of two primary components: first, the errors introduced by the Taylor expansion approximation in the derivation of the influence function, where high-order terms are neglected; and second, the errors arising from the new algorithm proposed in Section 3.3 in our manuscript for solving the inverse Hessian-vector product. We will conduct the error analysis in two parts accordingly.
>
> ### **1. Errors analysis for Taylor expansion approximation**
>
> Without loss of generality, we consider approximation error in Equation (6) of our submitted manuscript. In other words, we will analyze the error $\|\Delta\Theta(\epsilon) + \epsilon H_{\hat{\Theta}}^{-1}\nabla_{\Theta}\mathcal{L}((x, y); \hat{\Theta})\|.$
>
> The derivation below follows from [2], where we assume that $H_{\hat{\Theta}}$ is invertiable. As we discussed in our manuscript, this can be guaranteed if the second-order sufficient condition holds at $\hat{\Theta}$.
>
> Since $\hat{\Theta}_{new}(\epsilon)$ is an optimal solution to the perturbed loss function defined in Equation (5) in the submitted manuscript, we have
>
> $\nabla_{\Theta}R(\mathcal{Z}; \hat{\Theta}_{new}(\epsilon)) + $
>
> $\epsilon\nabla_{\Theta}\mathcal{L}((x, y); \hat{\Theta}_{new}(\epsilon)) = 0.$
>
>  Since $\hat{\Theta}_{new}(\epsilon) \approx \hat{\Theta}$ when $\epsilon$ is sufficiently small, it follows from the Taylor expansion that
>
> $$
> 0 = [\nabla_{\Theta}R(\mathcal{Z}; \hat{\Theta}) + \epsilon\nabla_{\Theta}\mathcal{L}((x, y); \hat{\Theta})] + [H_{\hat{\Theta}} + \epsilon\nabla_{\Theta}^2\mathcal{L}((x, y); \hat{\Theta})]\Delta\Theta(\epsilon) + o(\|\Delta\Theta(\epsilon)\|) .
> $$
>
>
> Since $\hat{\Theta}$ is an optimal solution to the loss function defined in Equation (3) in the submitted manuscript, we have
>
> $$
> \nabla_{\Theta}R(\mathcal{Z}; \hat{\Theta}) = 0.
> $$
>
> Therefore,
>
> $$
> \Delta\Theta(\epsilon) = -[H_{\hat{\Theta}} + \epsilon\nabla_{\Theta}^2\mathcal{L}((x, y); \hat{\Theta})]^{-1}(\epsilon\nabla_{\Theta}\mathcal{L}((x, y); \hat{\Theta}) + o(\|\Delta\Theta(\epsilon)\|).
> $$
>
>
>
> Since $\hat{\Theta}$ is an optimal solution to the loss function defined in Equation (3) in the submitted manuscript, $H_{\hat{\Theta}}$ is positive semidefinite. Therefore, the assumption that $H_{\hat{\Theta}}$ is invertiable implies that $H_{\hat{\Theta}}$ is positive definte. Therefore, we know that
>
> $$
> \Delta\Theta(\epsilon) = -H_{\hat{\Theta}}^{-1}(\epsilon\nabla_{\Theta}\mathcal{L}((x, y); \hat{\Theta}) + o(|\epsilon|)\|\Delta\Theta(\epsilon)\| + o(\|\Delta\Theta(\epsilon)\|).
> $$
>
>
>
> Therefore, as $\epsilon \to 0$,
>
> $$
> \|\Delta\Theta(\epsilon) + H_{\hat{\Theta}}^{-1}(\epsilon\nabla_{\Theta}\mathcal{L}((x, y); \hat{\Theta})\| = o(|\epsilon|) + o(\|\Delta\Theta(\epsilon)\|) \to 0.
> $$
>
>
>
>
> In our applications, we know that $\epsilon = O(1/n)$, where $n$ is the number of training samples. Therefore, $\epsilon$ should be very small and our approximation to $\Delta\Theta(\epsilon)$ by the influence function should be accurate for applications with a very large training datasets.
>
>
>
> ### **2. Errors analysis for our proposed algorithm**
>
> For our proposed Algorithm, the estimation errors analysis is as follows.
> For a given (approximate) solution $\widetilde{\Delta}$ to the linear system (12) in our manuscript, the error is defined as
> $$err(\widetilde{\Delta}) := \|\nabla^2_{\Theta}R(\mathcal{Z}; \widehat{\Theta})\widetilde{\Delta} - b\| = \|\nabla F(\widetilde{\Delta})\|,$$
> where the function $F(\cdot)$ is defined in (14) in the submitted manuscript. Therefore, the theoretical analysis of $err(\widetilde{\Delta})$ is equivalent to the error analysis of $\|\nabla F(\Delta_t)\|$ for the sequence $\{\Delta_t\}_{t \geq 1}$ generated by the optimization algorithm for solving the problem (9),(10), and (11) in the submitted manuscript.
>
> Since we use ADAM as a default optimizer for solving (9),(10),and (11), we analyze the error $\|\nabla F(\Delta_t)\|$ for the sequence $\{\Delta_t\}_{t \geq 1}$ generated by ADAM. It follows from [2] that ADAM can converge without modifications if the hyper-parameters are appropriately chosen (say the default choice $\beta_1 = 0.9$, $\beta_2 = 0.999$).
>
> Moreover, under reasonable assumptions (see [2] for more details), it holds that
> $$\min_{k_m \leq t \leq T} \mathbb{E}\|\nabla F(\Delta_t)\|_2 = \mathcal{O}(\log T/\sqrt{T}) = \widetilde{\mathcal{O}}(1/\sqrt{T}).$$
>
> Since for sufficiently large $T$, $\log T < T^q$ for any $q > 0$, we know we can achieve $$\min_{k_m \leq t \leq T} \mathbb{E}\|\nabla F(\Delta_t)\|_2 \leq \epsilon$$ for small $\epsilon > 0$ **in $\widetilde{\mathcal{O}}(\epsilon^{-2}) \approx O(\epsilon^{-2})$ iterations**. This proof also ensures the convergence of the algorithm proposed in Section 3.3.
>
> We will include these error analyses in the appendix of the revised version. Thank you for your comment!
>
> --------

---

> ### Author Response · Authors · 2024-11-24
> **Response to reviewer nWJj --- Part 5**
>
> >### Summary
>
> We thank you for your approval on our motivation, presentation, and effectiveness. We hope to address your concerns with:
>
> - Discussion about memory efficiency
> - Performance on generative tasks
> - Estimation errors analysis
> - Specific experimental setup
>
> We sincerely hope that our additional response could address your concerns. If so, we would greatly **appreciate your consideration in rasing the score.** If there are any remaining concerns, please let us know, and we will **continue to actively address your comments** and further improve our work.
>
> [1] Pearlmutter, B. A. (1994). Fast exact multiplication by the Hessian. Neural computation, 6(1), 147-160.
>
> [2] Zhang, Y., Chen, C., Shi, N., Sun, R., & Luo, Z. Q. (2022). Adam can converge without any modification on update rules. Advances in neural information processing systems, 35, 28386-28399. NIPS 2022.
>
> [3] Koh, P. W., & Liang, P. (2017, July). Understanding black-box predictions via influence functions. In International conference on machine learning (pp. 1885-1894). PMLR.
>
>
> [4] Chen, Y., Liu, Y., Chen, L., & Zhang, Y. (2021). DialogSum: A real-life scenario dialogue summarization dataset. arXiv preprint arXiv:2105.06762.
>
> [5] Yu, W., Yang, Z., Li, L., Wang, J., Lin, K., Liu, Z., ... & Wang, L. (2023). Mm-vet: Evaluating large multimodal models for integrated capabilities. arXiv preprint arXiv:2308.02490.

---

> > ### Comment · Reviewer_nWJj · 2024-11-26
> >
> > Thanks for the response. In your estimation errors analysis, is it dependent on the dimension d (rank r in LoRA)?

---

> > > ### Author Response · Authors · 2024-11-27
> > > **Response to Reviewer nWJj**
> > >
> > > Thank you for your response and question! In summary, the approximation error of LLMEraser is related to the number of trainable parameters (i.e., $d$ and $r$). However, when $T$ is sufficiently large and $\epsilon$ is sufficiently small, the approximation error of LLMEraser remains relatively small. As discussed in our estimation errors analysis, the approximation errors of LLMEraser arise from two sources: the error from the Taylor expansion and the error from the approximation of parameter changes introduced by the new algorithm in Section 3.3.
> > >
> > > Regarding the error analysis for our proposed algorithm, it is actually **independent of $d$ and the LoRA rank $r$**. In fact, as $T$ approaches infinity, the error $err (\widetilde{\Delta})$ **tends to zero**.
> > >
> > > For the errors introduced by the Taylor expansion approximation, **$(o(|\epsilon|)$ is independent of both $d$ and $r$**. For $o(\| \Delta \Theta(\epsilon) \|)$, the error is related to the parameter changes, and when $\epsilon$ is sufficiently small, the error is also relatively small.
> > >
> > > Overall, when the number of trainable parameters is large, a smaller $\epsilon$ and a larger number of iterations $T$ should be used to achieve a smaller error. In other words, as long as **$\epsilon$ is sufficiently small and $T$ is sufficiently large**, the **error in LLMEraser will remain relatively small** in general.
> > >
> > > In our response to Comment 2, the experimental results regarding different ranks of LoRA also demonstrate that, regardless of the rank setting used in LoRA, LLMEraser effectively reduces the negative impact of noisy data and achieves a significant utility gain, closely approximating the performance of retraining. This can be attributed to LLMEraser's relatively accurate estimation of the parameter changes compared to retraining.
> > >
> > > We sincerely hope that all of your concerns have been addressed and greatly appreciate your support. If you have any remaining questions, we are more than happy to engage further.

---

> > > ### Author Response · Authors · 2024-12-01
> > >
> > > Dear Reviewer nWJj,
> > >
> > > As the discussion phase comes to a close, we sincerely hope that our additional response has addressed your concerns. If so, we would greatly appreciate your consideration in raising the score. If there are any remaining concerns, we would be grateful if you could let us know, and we will make every effort to further refine and enhance our work.
> > >
> > > Best regards,
> > >
> > > The Authors of Paper 6232

---

> > > > ### Comment · Reviewer_nWJj · 2024-12-02
> > > >
> > > > Thanks for the author's further experimentation, clarification, and explanation. I've raised my score.

---

> > > > > ### Author Response · Authors · 2024-12-03
> > > > > **Thanks!**
> > > > >
> > > > > Thank you for your recognition of our work! We are truly encouraged and are committed to further refining our research. If you have any questions or suggestions, please don't hesitate to share them with us. We would be delighted to continue the discussion and enhance the paper. Once again, thank you for your time and efforts!

---

### Official Review · Reviewer_8XJc · 2024-11-02

**Soundness:** 2
**Presentation:** 3
**Contribution:** 2
**Rating:** 6
**Confidence:** 3

**Summary:**

This paper studies machine unlearning for parameter-efficient fine-tuning (PEFT) of large language models (LLM). The authors first propose a framework called LLMEraser for three different unlearning tasks: instance removal, query modification, and response correction. The core of LLMEraser is based on the influence function, based on which the algorithm can minimize the effect of the removed or modified samples. Some experiment results are presented to show the effectiveness of the proposed algorithm.

**Strengths:**

1) The paper shows the motivation and derivation of the proposed algorithm/framework.
2) The proposed algorithm demonstrates that it can effectively maintain good performance on different tasks, including recommendation and relation mining tasks.
3) The proposed algorithm is also shown to have high efficiency compared to the existing algorithms.

**Weaknesses:**

1) All the experiments show how well the models can perform after unlearning a certain amount of training samples. However, it seems the paper does not present or compare how effectively the algorithms can let the model "forget" those training samples, which is the original goal of the unlearning algorithm.
2) As PEFT is fine-tuning the model, it is unclear how to distinguish the influence of a sample when it is in the fine-tuning training set or the pre-training training set. When the sample was also used in the pre-training set, it is also unclear whether it is reasonable to require the PEFT to eliminate those
3) The experiment settings in the paper do not include the generative tasks of LLMs.
4) Some minor typos, such as "handling handle" near the end of page 9.

**Questions:**

1) How effectively can the proposed algorithm let the model forget the samples?
2) How effectively can the proposed algorithm maintain the model performance on generative tasks?
3) If a sample is also used in the pre-training, can it still be unlearned?
4) How can we tell whether the unlearning (or failure to unlearn) is on the PEFT module but not on the pre-trained model?

---

> ### Author Response · Authors · 2024-11-24
> **Response to reviewer 8XJc --- Part 1**
>
> We sincerely appreciate your positive feedback and thoughtful suggestions. Below, we provide detailed responses to address your remaining concerns.
>
> --------
> >### Comment 1 How effectively can the proposed algorithm let the model forget the samples?
>
> Thank you for your question! We will explain the effectiveness of unlearning through the following two points.
>
> - Many studies validate the effectiveness of unlearning through **adversarial attack experiments** [1-3]. First, corrupted instances are randomly introduced into the training dataset, causing a decline in the model’s performance. Unlearning techniques are then applied to correct the impact of these noisy data on the model. Ideally, an unlearning method should effectively mitigate the influence of noisy data and achieve performance comparable to that of retraining. Our evaluation on the QM task for LLMs and the RC task for MLLMs can be found in Tables 3, 4, and 5 of our manuscript. Overall, LLMEraser provides a substantial **utility gain** over the corrupted baseline, significantly mitigating the adverse effects of noisy data. Furthermore, its performance is comparable to that of Retrain, demonstrating its **effectiveness** in achieving data unlearning.
>
> - For the Instance Removal (IR) task, we have also included experiments **evaluated on the forgotten data**. We aim for the unlearning method to exhibit performance **close to Retrain**, essentially as if the model had never encountered removed samples. Here are the experimental results using TallRec as the backbone. The evaluation metric is AUC.
>
> |        | Original | Retrain | **LLMEraser (Ours)** |
> |:------:|:--------:|:-------:|:--------------------:|
> |  Test   |  0.6400  |  0.6357 |      **0.6319**      |
> | Forget |  0.6571  |  0.2571 |      **0.2714**      |
>
>
> We can observe that LLMEraser performs comparably to retraining on both the forgotten data and the test data. This demonstrates its ability to effectively unlearn data without compromising model utility, achieving performance on par with retraining.
>
> --------
> >### Comment 2 How can we tell whether the unlearning (or failure to unlearn) is on the PEFT module but not on the pre-trained model?
>
> Thank you for your suggestion! LLMEraser focuses on **unlearning domain-specific data exclusively used in PEFT**. Its application scenario arises when certain domain-specific data used for PEFT becomes unavailable. By leveraging influence functions, LLMEraser efficiently computes the parameter changes caused by removing these samples, eliminating the need for retraining.
>
> For the scenario you mentioned, where the same data is used for both pretraining and PEFT, we conducted the following experiments to explore its effects.
>
> - We first fine-tune the FLAN-T5 Base model on the DIALOGSUM [4] dataset to simulate the pretraining process (Full FT) due to the time constraint. Subsequently, a subset of this data is used for PEFT (Full FT + PEFT), meaning this subset is utilized in both pretraining and PEFT. Following PEFT, we apply LLMEraser to unlearn this subset (Full FT + PEFT + Ours). The evaluation results on the test data are shown in the table below.
>
> |                           |  ROUGE-1  |  ROUGE-2 |  ROUGE-L  |
> |:-------------------------:|:---------:|:--------:|:---------:|
> |            Base           |   20.55   |   5.63   |   20.05   |
> |          Full FT          |   30.47   |   10.92   |   26.46   |
> |       Full FT + PEFT      |   36.79   |   10.59  |   29.65   |
> | **Full FT + PEFT + Ours** | **30.28** | **8.22** | **25.65** |
>
> We observe that LLMEraser effectively unlearns the information used in PEFT. The performance of Full FT + PEFT + Ours is comparable to that of Full FT. This is because LLMEraser utilizes the influence function to **compute the gradient information of the PEFT adapter's parameters** and subsequently calculates the corresponding parameter changes.

---

> ### Author Response · Authors · 2024-11-24
> **Response to reviewer 8XJc --- Part 2**
>
> - On LLAMA 2-7B, we first perform PEFT using the Alpaca-LoRA dataset for instruction tuning, simulating the pretraining process (Simulated FT). Subsequently, a subset of this data is reused along with new LoRA modules for additional PEFT (Simulated Pretraining + PEFT). This subset is thus utilized in both the simulated pretraining and PEFT stages. Finally, we apply LLMEraser to unlearn this subset (Simulated Pretraining + PEFT + Ours). The evaluation results on the forgotten data are presented in the table below.
>
> |                  |  ROUGE-1  |  ROUGE-2  |  ROUGE-L  |
> |:----------------------:|:---------:|:---------:|:---------:|
> |          Simulated Pretraining          |   39.83   |   14.67   |   29.23   |
> |       Simulated Pretraining + PEFT      |   42.97   |   16.99   |   30.77   |
> | **Simulated Pretraining + PEFT + Ours** | **40.13** | **15.03** | **27.63** |
>
>
> In summary, we use full fine-tuning and PEFT to simulate the pretraining process, thereby replicating scenarios where data appears in both the pretraining and PEFT stages. For LLMEraser, it leverages the influence function to **compute the parameter changes in the PEFT adapter**. In other words, LLMEraser relies on the gradient information of the old adapter and is capable of unlearning information of training samples learned in old adapters and modifying the adapter's parameters.
>
> Please let us know if this answers your question; we are happy to provide further explanation if needed.
>
> --------
>
> > ### Comment 3 If a sample is also used in the pre-training, can it still be unlearned?
>
> Thank you for your comment! Please refer to the relevant experiments discussed in our response to Comment 1. We would like to highlight that LLMEraser is designed to unlearn the domain-specific data information used in PEFT and to efficiently update the adapter’s parameters. In this setting, LLMEraser **specifically targets the information learned by the adapter**. Notably, the gradient information utilized by the influence function is also derived solely **from the adapter’s parameters**.
>
> Please let us know if this answers your question; we are happy to provide further explanation if needed.
>
> --------
>
> >### Comment 4 Performance on generative tasks of LLMs
>
> Thanks for the comment! Generative tasks is highly significant for LLMs. We **have added experiments on various text generation tasks**, including dialogue summarization and question-answering tasks, covering Instance Removal (IR), Query Modification (QM), and Response Correction (RC) tasks.
>
> ### **1. Results of the Instance Removal (IR) task**
>
> For **Instance Removal (IR) task**, we conduct experiments on Alpaca-LoRA for the instruction tuning task. The backbone of LLM is LLaMA 2-7B. We remove 5% of the training samples and evaluate the performance of the original model, retrained model, and LLMEraser on the deleted data using ROUGE scores as the evaluation metric. Here are the experimental results.
>
> |    Method    |  ROUGE-1  |  ROUGE-2 |  ROUGE-L  |
> |:-------------:|:---------:|:---------:|:---------:|
> |    Original   |   44.23   |   27.09   |   34.88   |
> |    Retrain    |   39.84   |   20.40   |   31.78   |
> | **LLMEraser (Ours)** | **39.55** | **20.17** | **31.63** |
>
>
> We can oberserve that LLMEraser closely matches the performance of Retrain. This is attributed to its direct estimation of parameter changes between the retrained model and the original model, enabling accurate calculations of these changes.

---

> ### Author Response · Authors · 2024-11-24
> **Response to reviewer 8XJc --- Part 3**
>
> ### **2. Results of the Query Modification (QM) and Response Correction (RC) tasks**
>
> For **Query Modification (QM) and Response Correction (RC) tasks**, we have added experimental results on the DIALOGSUM [4] dataset for dialogue summarization. The backbone of the LLM is FLAN-T5 Base due to time constraints. We will include additional experimental results in the future. Specifically, we apply perturbations to 50% of the samples in the dialogue and 20% of the samples in the summary, and use LLMEraser to correct the corrupted data. Here are the experimental results.
>
> |                         |      Method         | ROUGE-1  | ROUGE-2   | ROUGE-L  |
> |-------------------------|----------------------|-----------|-----------|-----------|
> |                         | Retrain            | 36.59   | 12.78   | 29.56   |
> | 50% Dialogue Distortion | Corrupted            | 34.09     | 11.61     | 26.66     |
> |                         | **LLMEraser (Ours)** | **35.92** | **12.22** | **28.98** |
> |                         | Retrain            | 36.59   | 12.78   | 29.56   |
> | 20% Summary Distortion  | Corrupted            | 35.72     | 11.67     | 29.09     |
> |                         | **LLMEraser (Ours)** | **36.34** | **12.49** | **29.45** |
>
> For MLLM, we conduct experiment on the mm-vet-v2 dataset for integrated capability evaluation task [6]. The data format of mm-vet-v2 is actually in the form of text-based question-answer pairs. The backbone of MLLM is LLaVA 1.5-7B. Specifically, we randomly select 80% of mm-vet-v2 samples as training set, and employ the left 20% samples for evaluation. We apply perturbations to 50% of the training samples and evaluate the performance of the retrained model, corrupted model and LLMEraser on the testing set,  where LLMEraser corrects the corrupted data. Here are the experimental results on the **Query Modification (QM)** task, where we utilize "rec, gen, ocr, spat, know" capacities  for evaluation, and report the average results. (All Experimental scores are calculated with gpt-4-turbo by following [6].)
>
>
> |                      |    rec   |    gen   |    ocr   |   spat  |   know  |   Average |
> |:--------------------:|:--------:|:--------:|:--------:|:--------:|:--------:|:--------:|
> |        Retrain       |  37.8 |  28.9  |  28.3 |  37.7  |  21.6  |    30.9 |
> |       Corrupted      |   29.4   |   23.0   |   20.7   |   34.1   |   14.1   |   24.3   |
> | **LLMEraser (Ours)** | **33.1** | **27.0** | **21.5** | **37.2** | **17.5**| **27.3** |
>
> LLMEraser provides a substantial utility improvement to the model compared to the corrupted baseline, effectively mitigating the negative impact of noisy data. On the LLM dialogue summarization task, LLMEraser achieves an average improvement of **6.45%** on QM tasks and **3.34%** on RC tasks compared to the corrupted baseline. For MLLM QA tasks, LLMEraser achieves an average improvement of **12.35%** on QM task. Furthermore, LLMEraser’s performance is close to that of Retrain, highlighting its effectiveness in correcting inaccurate information.
>
>
> In the revised version, we will include these experimental results and implementation details.
>
>
> --------
> >### Comment 5 How to distinguish the influence of a sample when it is in the fine-tuning training set or the pre-training training set?
>
> Thanks for asking! How to distinguish the influence of a sample when it is part of the fine-tuning training set or the pre-training training set is an interesting question, and some existing works [5] have explored this topic. In LLMEraser's experimental setup, it is not necessary to distinguish the impact of samples during the pre-training or fine-tuning stages. This is because LLMEraser **solely utilizes the gradient information from the PEFT adapter** to update the adapter's parameters, thereby only forgetting the knowledge learned by the adapter. In the future, we plan to further investigate this topic, as we believe there is potential for technological solutions to address it.
>
> --------
> >### Comment 6 Some minor typos
>
> Thank you for your thorough review of our work! We sincerely apologize for the oversight and will address the presentation issues with great care in the revised version. Furthermore, we will thoroughly review the presentation across the entire manuscript.

---

> ### Author Response · Authors · 2024-11-24
> **Response to reviewer 8XJc --- Part 4**
>
> >### Summary
>
> We hope to address your concerns with:
>
> - Unlearning setting of data used in both pretraining and PEFT.
> - Unlearning effectiveness
> - Performance on generative tasks
>
> We sincerely **appreciate your support and positive feedback** on the presentation, novelty, and effectiveness of our work. We appreciate it if you **could reconsider your evaluation** if some concerns are addressed. Thanks!
>
> [1] Jiancan Wu, Yi Yang, Yuchun Qian, Yongduo Sui, Xiang Wang, and Xiangnan He. GIF: A general graph unlearning strategy via influence function. In WWW, pp. 651–661. ACM, 2023a.
>
> [2] Saemi Moon, Seunghyuk Cho, and Dongwoo Kim. Feature unlearning for pre-trained gans and vaes. In AAAI, pp. 21420–21428. AAAI Press, 2024.
>
> [3] Sungmin Cha, Sungjun Cho, Dasol Hwang, Honglak Lee, Taesup Moon, and Moontae Lee. Learning to unlearn: Instance-wise unlearning for pre-trained classifiers. In AAAI, pp. 11186–11194. AAAI Press, 2024.
>
> [4] Yulong Chen, Yang Liu, Liang Chen, and Yue Zhang. Dialogsum: A real-life scenario dialogue summarization dataset. In ACL/IJCNLP (Findings), volume ACL/IJCNLP 2021 of Findings of ACL, pp. 5062–5074. Association for Computational Linguistics, 2021.
>
> [5] Weichao Zhang, Ruqing Zhang, Jiafeng Guo, Maarten de Rijke, Yixing Fan, and Xueqi Cheng. Pretraining data detection for large language models: A divergence-based calibration method. In EMNLP, pp. 5263–5274. Association for Computational Linguistics, 2024a.
>
> [6] Weihao Yu, Zhengyuan Yang, Linjie Li, Jianfeng Wang, Kevin Lin, Zicheng Liu, Xinchao Wang, and Lijuan Wang. Mm-vet: Evaluating large multimodal models for integrated capabilities. In ICML. OpenReview.net, 2024.

---

> ### Author Response · Authors · 2024-12-01
>
> Dear Reviewer 8XJc,
>
> As the discussion phase comes to a close, we sincerely hope that our additional response has addressed your concerns. If so, we would greatly appreciate your consideration in raising the score. If there are any remaining concerns, we would be grateful if you could let us know, and we will make every effort to further refine and enhance our work.
>
> Best regards,
>
> The Authors of Paper 6232

---

> > ### Comment · Reviewer_8XJc · 2024-12-01
> >
> > Thanks for the clarification from the authors. The rebuttal mitigates some of my concerns. So I will raise my score.
> >
> > However, the experiments mainly focus on adversarial attack experiments to test how well the proposed method can erase the effect of removing samples, which may be an extreme case in practice. A more common scenario is to erase natural, benign samples (e.g., for privacy concerns instead of adversarial concerns), just like the first part of experiments in [2] mentioned by the authors.

---

> > > ### Author Response · Authors · 2024-12-03
> > > **Thanks!**
> > >
> > > Thank you for your recognition of our work!
> > >
> > > Thank you for your valuable suggestions regarding the experimental setup! In response to Comment 1, we have added additional experimental results on forgot data, specifically focusing on the removal of normal samples. In fact, LLMEraser is a unified unlearning framework capable of removing any training data used in PEFT, regardless of the data type. We believe these applications would serve as interesting future directions for applying LLMEraser to unlearning sensitive information. Thank you again for your insightful suggestions!
> > >
> > > We are truly encouraged and are committed to further refining our research. If you have any questions or suggestions, please don't hesitate to share them with us. We would be delighted to continue the discussion and enhance the paper. Once again, thank you for your time and efforts!

---

### Official Review · Reviewer_8gYQ · 2024-11-04

**Soundness:** 3
**Presentation:** 4
**Contribution:** 3
**Rating:** 8
**Confidence:** 3

**Summary:**

This paper introduce a novel instance-wise unlearning method that utilizes influence functions to adjusting parameters for unlearning tasks. Specifically, the method can remove specific sample from the training dataset, adjust input tokens in user queries, as well as correct model response. Experimental results show that the method can perform unlearning tasks with high accuracy while ensuring efficiency.

**Strengths:**

1. The problem is important and interesting. The three unlearning tasks are practical and meaningful in real-world settings.

2. The unlearning approach edits model parameters based on influence functions, which avoids finetuning or re-training the models

3. The evaluations are comprehensive.

4. The writing is clear and easy to follow.

**Weaknesses:**

My concern is mainly about evaluation datasets.

1. Although the paper claims to address privacy concerns in LLMs, the authors primarily evaluated the method using tabular datasets for recommendation tasks and a multimodal dataset for a classification task. While these tasks are representative, they are relatively simple and straightforward. Although the authors applied prompt engineering to those data, the training data are still very similar.

2. In real AI applications, LLMs are used for text generation more often, and the training data consists of documents from various sources and may contain sensitive information, such as bank accounts, addresses, and SSNs. Thus, this paper lacks evaluations in more practical and meaningful scenarios that would better demonstrate its effectiveness in addressing privacy concerns in real-world settings.

**Questions:**

see weakness. Also, how is the performance of the method on text-generation tasks? Could you evaluate your approach on some QA datasets?

---

> ### Author Response · Authors · 2024-11-24
> **Response to reviewer 8gYQ --- Part 1**
>
> We appreciate your efforts and insightful comments! To address your concerns, we provide detailed responses below.
>
> >### Comment 1 How is the performance of the method on text-generation tasks? Could you evaluate your approach on some QA datasets?
>
> --------
>
>
> Thank you for your comment! We believe that text generation is essential for large language models. Therefore, we have added experiments on various types of text generation tasks.
>
> ### **1. Results of the Instance Removal (IR) task**
>
> For **Instance Removal (IR) task**, we conducted experiments on Alpaca-LoRA for the instruction tuning task. The backbone of LLM is LLaMA 2-7B. We removed 5% of the training samples and evaluate the performance of the original model, retrained model, and LLMEraser on the deleted data using ROUGE scores as the evaluation metric. Here are the experimental results.
>
> |    Method    |  ROUGE-1  |  ROUGE-2 |  ROUGE-L  |
> |:-------------:|:---------:|:---------:|:---------:|
> |    Original   |   44.23   |   27.09   |   34.88   |
> |    Retrain    |   39.84   |   20.40   |   31.78   |
> | **LLMEraser** | **39.55** | **20.17** | **31.63** |
>
>
> We can observe that LLMEraser closely matches the performance of Retrain. This is attributed to its direct estimation of parameter changes between the retrained model and the original model, enabling accurate calculations of these changes.
>
> ### **2. Results of the Query Modification (QM) and Response Correction (RC) tasks**
>
> For **Query Modification (QM) and Response Correction (RC) tasks**, we have added experimental results on the DIALOGSUM [1] dataset for dialogue summarization. The backbone of the LLM is FLAN-T5 Base due to time constraints. We will include additional experimental results in the future. Specifically, we applied perturbations to 50% of the samples in the dialogue and 20% of the samples in the summary, and used LLMEraser to correct the corrupted data. Here are the experimental results.
>
> |                         |      Method         | ROUGE-1  | ROUGE-2   | ROUGE-L  |
> |-------------------------|----------------------|-----------|-----------|-----------|
> |                         | Retrain            | 36.59   | 12.78   | 29.56   |
> | 50% Dialogue Distortion | Corrupted            | 34.09     | 11.61     | 26.66     |
> |                         | **LLMEraser (Ours)** | **35.92** | **12.22** | **28.98** |
> |                         | Retrain            | 36.59   | 12.78   | 29.56   |
> | 20% Summary Distortion  | Corrupted            | 35.72     | 11.67     | 29.09     |
> |                         | **LLMEraser (Ours)** | **36.34** | **12.49** | **29.45** |
>
> For MLLM, we conduct experiment on the mm-vet-v2 dataset for integrated capability evaluation task [2]. The data format of mm-vet-v2 is actually in the form of text-based question-answer pairs. The backbone of MLLM is LLaVA 1.5-7B. Specifically, we randomly select 80% of mm-vet-v2 samples as training set, and employ the left 20% samples for evaluation. We applied perturbations to 50% of the training samples and evaluated the performance of the retrained model, corrupted model and LLMEraser on the testing set,  where LLMEraser corrects the corrupted data. Here are the experimental results on the **Query Modification (QM)** task, where we utilized "rec, gen, ocr, spat, know" capacities  for evaluation, and reported the average results. (All Experimental scores are calculated with gpt-4-turbo by following [2].)
>
>
> |                      |    rec   |    gen   |    ocr   |   spat  |   know  |   Average |
> |:--------------------:|:--------:|:--------:|:--------:|:--------:|:--------:|:--------:|
> |        Retrain       |  37.8 |  28.9  |  28.3 |  37.7  |  21.6  |    30.9 |
> |       Corrupted      |   29.4   |   23.0   |   20.7   |   34.1   |   14.1   |   24.3   |
> | **LLMEraser (Ours)** | **33.1** | **27.0** | **21.5** | **37.2** | **17.5**| **27.3** |
>
> LLMEraser provides a substantial utility improvement to the model compared to the corrupted baseline, effectively mitigating the negative impact of noisy data. On the LLM dialogue summarization task, LLMEraser achieves an average improvement of **6.45%** on QM tasks and **3.34%** on RC tasks compared to the corrupted baseline. For MLLM QA tasks, LLMEraser achieves an average improvement of **12.3%** on QM task. Furthermore, LLMEraser’s performance is close to that of Retrain, highlighting its effectiveness in correcting inaccurate information.
>
> In the revised version, we have included the experimental results and related analysis for these generative tasks. More case studies on the supplementary experiments will be presented in the appendix of the revised version. We will include more experimental results in future versions. More details about the experiments can be found in the appendix of the revised version.

---

> ### Author Response · Authors · 2024-11-24
> **Response to reviewer 8gYQ --- Part 2**
>
> >### Comment 2 Discussion about the application
>
>
> Thanks fou your insightful comment! LLMEraser is designed to unlearn domain-specific data, enabling the forgetting or correction of data regardless of its type. In practice, many recommendation datasets are related to user behavior, inherently involving some level of user privacy. If you have relevant privacy datasets, we would be happy to include additional experiments.
> We believe these applications would serve as interesting future directions for applying LLMEraser to unlearning sensitive information. Thank you for your suggestion!
>
>
> --------
>
>
> >### Summary
>
> We hope to address your concerns with:
>
> - Model performance on generative tasks.
> - Discussion about the application.
>
> We also appreciate your support and positive feedback on our presentation, novelty, and effectiveness.
>
> Overall, we hope our response justifies the generalization ability of LLMEraser, showcasing its generalization ability across various tasks, and we would appreciate your reconsideration. We would like to highlight that our main contribution lies in proposing a unified framework for parameter-efficient unlearning in large language models. Additionally, we provide a detailed taxonomy for unlearning tasks, enabling effective data forgetting or correction across various types of tasks. Thank you for your efforts again!
>
> [1] Yulong Chen, Yang Liu, Liang Chen, and Yue Zhang. Dialogsum: A real-life scenario dialogue summarization dataset. In ACL/IJCNLP (Findings), volume ACL/IJCNLP 2021 of Findings of ACL, pp. 5062–5074. Association for Computational Linguistics, 2021.
>
>
> [2] Weihao Yu, Zhengyuan Yang, Linjie Li, Jianfeng Wang, Kevin Lin, Zicheng Liu, Xinchao Wang, and Lijuan Wang. Mm-vet: Evaluating large multimodal models for integrated capabilities. In ICML. OpenReview.net, 2024.

---

> > ### Comment · Reviewer_8gYQ · 2024-11-24
> >
> > Thank you for your efforts. I have adjust my score. Good luck.

---

> > > ### Author Response · Authors · 2024-11-25
> > > **Thanks!**
> > >
> > > Thank you for your recognition of our work and rebuttal. We are truly encouraged and plan to further refine our research. If you have any questions or suggestions, please feel free to share them with us. We would be delighted to continue discussing and improving the paper. Once again, thank you for your efforts!

---

### Official Review · Reviewer_W27P · 2024-11-04

**Soundness:** 4
**Presentation:** 3
**Contribution:** 2
**Rating:** 8
**Confidence:** 4

**Summary:**

This paper suggest a unified and feasible way to address multiple unlearning scenarios under a single framework. It first generalizes three unlearning objectives. Then, proposes to use influence-functions to approximate the influence of datapoints requested to be unlearned. It then proposed to use PEFT to avoid full finetuning of the model, by training each adapter separately.

**Strengths:**

The paper unifies three unlearning objective under a single simple framework.
The paper directly addresses the complexity concerns the framework seems to have in two simple ways: using fast Hessian-vector product, and using mini-batches. It then provides a convincing argument for why such problem formulation would converge using SGD.

**Weaknesses:**

The paper is a bit hard to read. I would suggest putting most equations in the appendix and focus on high-level explanation in the body of the paper. It is unclear exactly what exactly how adapters are used in Algorithm 1. The evaluation is a bit limited, consider comparing this method performance to other unlearning approaches. Evaluating efficiency in seconds can be a nice addition but the main evaluation should compare an asymptotic running-time complexity.

**Questions:**

In the limitation section is mentioned a limited due to "first-order Taylor expansion", but I thought influence functions rely on second order expansion, which one is it?

---

> ### Author Response · Authors · 2024-11-24
> **Response to Reviewer W27P --- Part 1**
>
> Response to reviewer W27P
>
> We sincerely appreciate your positive feedback and thoughtful suggestions! Below, we provide responses to address your remaining concerns in detail.
>
> --------
>
> > ### Comment 1 Reorganize the manuscript
>
> Thanks for your insightful comments! We are working on revising the presentation of the paper. In particular, we will revise the presentation of the technical parts according to your suggestion. We will include more high-level explanations and case studies to help readers better understand our intentions and methods. Additionally, we will move some of the equations to the appendix. We have also provided a detailed explanation of how LLMEraser works in our response to Comment 2.
>
> --------
> >  ### Comment 2 How adapters are used in Algorithm 1
>
> Thank you for your question! LLMEraser focuses on unlearning domain-specific data and updating the parameters of the PEFT adapters. As shown in Figure 2 of our manuscript, the old adapter is obtained through PEFT on domain-specific data. When there is a need to delete or correct certain data from the domain-specific data, i.e., when an unlearning request arrives, LLMEraser utilizes influence functions to compute the parameter changes caused by such request. After calculating the parameter changes, these changes are added to the parameters of the old adapter, resulting in the new adapter parameters—essentially the unlearned model parameters. The **model** mentioned in the Algorithm 1 refers to the old adapter. We leverage its gradient information to compute b and subsequently compute the parameter changes to update its parameters. This clarification will be addressed in the revised version.
> In summary, the **workflow of LLMEraser** is as follows:
>
> - Use domain-specific data with PEFT to obtain the old adapter.
> - Receive the unlearning request.
> - LLMEraser utilizes influence functions to compute the changes in model parameters caused by the unlearning request.
> - Add the computed parameter changes to the old adapter's parameters to obtain the unlearned model parameters.
>
> We will include these explanation in the revised version. Thank you for your suggestion!
>
> --------
> > ### Comment 3 More experimental evaluation
>
> Thanks for the comment! We have added experimental results for LLMEraser on additional generative tasks.
>
> ### **1. Results of the Instance Removal (IR) task**
>
> For **Instance Removal (IR) task**, we conduct experiments on Alpaca-LoRA for the instruction tuning task. The backbone of LLM is LLaMA 2-7B. We remove 5% of the training samples and evaluate the performance of the original model, retrained model, and LLMEraser on the deleted data using ROUGE scores as the evaluation metric. Here are the experimental results.
>
> |    Method    |  ROUGE-1  |  ROUGE-2  |  ROUGE-L  |
> |:-------------:|:---------:|:---------:|:---------:|
> |    Original   |   44.23   |   27.09   |   34.88   |
> |    Retrain    |   39.84   |   20.40   |   31.78   |
> | **LLMEraser** | **39.55** | **20.17** | **31.63** |
>
>
> We can observe that LLMEraser closely matches the performance of Retrain. This is attributed to its direct estimation of parameter changes between the retrained model and the original model, enabling accurate calculations of these changes.

---

> > ### Comment · Reviewer_W27P · 2024-11-26
> >
> > Thank you for your efforts. I have adjust my score. Good luck.

---

> > > ### Author Response · Authors · 2024-11-27
> > > **Thanks!**
> > >
> > > Thank you for your recognition of our work and rebuttal. Your positive feedback and constructive suggestions have truly encouraged us. If you have any additional questions or suggestions, please feel free to share them with us. We would be delighted to continue discussing and improving the paper. Once again, thank you for your valuable time and efforts in reviewing our submission!

---

> ### Author Response · Authors · 2024-11-24
> **Response to Reviewer W27P --- Part 2**
>
> ### **2. Results of the Query Modification (QM) and Response Correction (RC) tasks**
>
> For **Query Modification (QM) and Response Correction (RC) tasks**, we have added experimental results on the DIALOGSUM [5] dataset for dialogue summarization. The backbone of the LLM is FLAN-T5 Base due to time constraints. We will include additional experimental results in the future. Specifically, we apply perturbations to 50% of the samples in the dialogue and 20% of the samples in the summary, and use LLMEraser to correct the corrupted data. Here are the experimental results.
>
> |                         |      Method         | ROUGE-1  | ROUGE-2   | ROUGE-L  |
> |-------------------------|----------------------|-----------|-----------|-----------|
> |                         | Retrain            | 36.59   | 12.78   | 29.56   |
> | 50% Dialogue Distortion | Corrupted            | 34.09     | 11.61     | 26.66     |
> |                         | **LLMEraser (Ours)** | **35.92** | **12.22** | **28.98** |
> |                         | Retrain            | 36.59   | 12.78   | 29.56   |
> | 20% Summary Distortion  | Corrupted            | 35.72     | 11.67     | 29.09     |
> |                         | **LLMEraser (Ours)** | **36.34** | **12.49** | **29.45** |
>
> For MLLM, we conduct experiment on the mm-vet-v2 dataset for integrated capability evaluation task [6]. The data format of mm-vet-v2 is actually in the form of text-based question-answer pairs. The backbone of MLLM is LLaVA 1.5-7B. Specifically, we randomly select 80% of mm-vet-v2 samples as training set, and employ the left 20% samples for evaluation. We apply perturbations to 50% of the training samples and evaluate the performance of the retrained model, corrupted model and LLMEraser on the testing set,  where LLMEraser corrects the corrupted data. Here are the experimental results on the **Query Modification (QM)** task, where we utilize "rec, gen, ocr, spat, know" capacities  for evaluation, and report the average results. (All Experimental scores are calculated with gpt-4-turbo by following [6].)
>
> |                      |    rec   |    gen   |    ocr   |   spat  |   know  |   Average |
> |:------------:|:--------:|:--------:|:--------:|:--------:|:--------:|:--------:|
> |        Retrain       |  37.8 |  28.9  |  28.3 |  37.7  |  21.6  |    30.9 |
> |       Corrupted      |   29.4   |   23.0   |   20.7   |   34.1   |   14.1   |   24.3   |
> | **LLMEraser (Ours)** | **33.1** | **27.0** | **21.5** | **37.2** | **17.5**| **27.3** |
>
> LLMEraser provides a substantial utility improvement to the model compared to the corrupted baseline, effectively mitigating the negative impact of noisy data. On the LLM dialogue summarization task, LLMEraser achieves an average improvement of **6.45%** on QM tasks and **3.34%** on RC tasks compared to the corrupted baseline. For MLLM QA tasks, LLMEraser achieves an average improvement of **12.35%** on QM task. Furthermore, LLMEraser’s performance is close to that of Retrain, highlighting its effectiveness in correcting inaccurate information.
>
> As mentioned in our paper, contemporary **approximate unlearning** methods are mostly limited to Instance Removal (IR) tasks and are not well-suited for Query Modification (QM) and Response Correction (RC) tasks. On the other hand, **exact unlearning** cannot avoid retraining and significantly impacts model performance. Many LLM unlearning methods cannot be evaluated on QM and RC tasks. We chose Retrain, Gradient Ascent, and E2URec [4] as baselines for Instance Removal (IR) task. In the future, we will explore and follow up on other unlearning  methods. Thank you for your valuable suggestions!
>
>
>
> --------
>
> >### Comment 4 Evaluating efficiency and running-time complexity
>
> Thanks for the insightful comment! We have added runtime experiments for the TallRec backbone and updated the evaluation results in Section 4.4. We will include these experimental results in future versions as well. Here are the execution times in the QM task using LLaRA as the backbone.
>
> |          |     Retrain     |       SISA      |    RecEraser    |  LLMEraser (Ours) |
> |:--------:|:---------------:|:---------------:|:---------------:|:-----------------:|
> | Time (s) | $5.4\times10^4$ | $1.8\times10^4$ | $2.0\times10^4$ | **$1.4 \times 10^3$** |
>
> For TallRec, here are the execution times in the IR task.
>
> |          |     Retrain     | Gradient Ascent |     E2URec    |   LLMEraser (Ours)  |
> |:--------:|:---------------:|:--------------:|:-------------:|:-------------------:|
> | Time (s) | $5.6\times10^3$ | $2.3\times10^3$ | $2.4\times10^2$ | **$4.9\times 10^1$** |

---

> ### Author Response · Authors · 2024-11-24
> **Response to Reviewer W27P --- Part 3**
>
> Our method transforms the computation of the **inverse Hessian-vector product** into the problem of solving for the **Hessian-vector product**, enabling efficient resolution through **mini-batch** algorithms. The Hessian-vector product, if computed directly via the full Hessian matrix multiplication, would have a time and space complexity of $O(p^2)$. However, using **HVP** (Hessian-free methods), we **avoid the explicit computation and storage of the Hessian matrix**, reducing both time and space complexity to **$O(p)$** [3]. By further leveraging mini-batch optimization for equation 12 in the manuscript, LLMEraser achieves a space complexity of **$O(p)$**, ensuring its scalability.
>
> Regarding GPU memory usage, we have measured the GPU utilization of the LLaRA backbone with LoRA rank r sets to 8, 16, and 32. Here are the statistical information and the experimental results (with memory usage measured in megabytes (MB)).
>
> |     Method    |     LoRA r = 8     |     LoRA r = 16    |     LoRA r = 32    |
> |:-------------:|:---------:|:---------:|:---------:|
> |    Retrain    |   33040 MB |   33868 MB  |   34128  MB |
> |      SISA     |   33040 MB  |   33868 MB  |   34128  MB |
> | **LLMEraser** | **30760 MB** | **31386 MB** | **31834 MB** |
>
>
> For the **asymptotic running-time complexity**, methods like retrain or SISA, and gradient ascent require retraining or fine-tuning the PEFT adapter. Their time complexity is tied to the training complexity of the original adapter. In contrast, our method, LLMEraser, eliminates the need for retraining or tuning the adapter. Instead, it computes parameter updates directly and efficiently. The new algorithm we proposed in Section 3.3 leverages Hessian-Vector Product (HVP) to accelerate computation, reducing the time complexity from $\mathcal{O}\left(p^2\right)$ to $\mathcal{O}(p)$ [3], where p is the number of trainable parameters. Below, we provide an analysis of the **convergence** of our new algorithm.
>
> For a given (approximate) solution $\widetilde{\Delta}$ to the linear system (12) in our manuscript, the error is defined as
> $$err(\widetilde{\Delta}) := \|\nabla^2_{\Theta}R(\mathcal{Z}; \widehat{\Theta})\widetilde{\Delta} - b\| = \|\nabla F(\widetilde{\Delta})\|,$$
> where the function $F(\cdot)$ is defined in (14) in the submitted manuscript. Therefore, the theoretical analysis of $err (\widetilde{\Delta})$ is equivalent to the error analysis of $\|\nabla F(\Delta_t)\|$ for the sequence $\{\Delta_t\}_{t \geq 1}$ generated by the optimization algorithm for solving the problem (9),(10),and (11) in the submitted manuscript.
>
> Since we use ADAM as a default optimizer for solving (9),(10),and (11), we analyze the error $\|\nabla F(\Delta_t)\|$ for the sequence $\{\Delta_t\}_{t \geq 1}$ generated by ADAM. It follows from [1] that ADAM can converge without modifications if the hyper-parameters are appropriately chosen (say the default choice $\beta_1 = 0.9$, $\beta_2 = 0.999$).
>
> Moreover, under reasonable assumptions (see [1] for more details), it holds that
> $$\min_{k_m \leq t \leq T} \mathbb{E}\|\nabla F(\Delta_t)\|_2 = \mathcal{O}(\log T/\sqrt{T}) = \widetilde{\mathcal{O}}(1/\sqrt{T}).$$
>
> Since for sufficiently large $T$, $\log T < T^q$ for any $q > 0$, we know we can achieve $$\min_{k_m \leq t \leq T} \mathbb{E}\|\nabla F(\Delta_t)\|_2 \leq \epsilon$$ for small $\epsilon > 0$ **in $\widetilde{\mathcal{O}}(\epsilon^{-2}) \approx O(\epsilon^{-2})$ iterations**. This ensures the convergence of the algorithm proposed in Section 3.3.
>
>
> Please let us know if this addresses your question. We are happy to provide additional explanation if required.

---

> ### Author Response · Authors · 2024-11-24
> **Response to Reviewer W27P --- Part 4**
>
> > ### Comment 5 Discussion about the first-order Taylor expansion
>
> Thanks for asking! For the problem (7) in our paper, the optimal solution is given as $\widehat{\Theta_\delta}(\epsilon)$. We can obtain that:
> $\nabla_\Theta R(\mathcal{Z}; \widehat{\Theta_\delta})+\epsilon \nabla_\Theta(\mathcal{L}((x+\delta_x, y+\delta_y) ; \widehat{\Theta_\delta})-\epsilon \mathcal{L}((x, y) ; \widehat{\Theta_\delta}))=0$.
> Since the perturbation is relatively small, we can perform the Taylor expansion  at $\widehat\Theta$. Thus, we have:
> $\nabla_\Theta R(\mathcal{Z}; \widehat{\Theta})+\epsilon \nabla_\Theta(\mathcal{L}((x+\delta_x, y+\delta_y) ; \widehat{\Theta})-\epsilon \mathcal{L}((x, y) ; \widehat{\Theta})) +(\widehat{\Theta_\delta}(\epsilon) - \widehat{\Theta})[ \nabla_\Theta^2 R(\mathcal{Z}; \widehat{\Theta})+\epsilon \nabla_\Theta^2(\mathcal{L}((x+\delta_x, y+\delta_y) ; \widehat{\Theta})-\epsilon \mathcal{L}((x, y) ; \widehat{\Theta}))] + ... =0$.
>
> Since $\widehat{\Theta}$ is the optimal solution for problem (3) in our paper, by taking $\nabla_\Theta R(\mathcal{Z}; \widehat{\Theta})=0$ and ignoring high-order terms, we can approximate the parameter changes $\Delta\Theta = \widehat{\Theta_\delta}(\epsilon) - \widehat{\Theta}$ as equation (8) in our paper. This is consistent with the derivation in [2].
>
> The form of equation (8) indeed involves second-order gradients. However, this term effectively corresponds to the first-order term in the Taylor expansion because $ \nabla_\Theta R(\mathcal{Z}; \widehat{\Theta_\delta}) + \epsilon \nabla_\Theta(\mathcal{L}((x+\delta_x, y+\delta_y); \widehat{\Theta_\delta}) - \mathcal{L}((x, y); \widehat{\Theta_\delta})) = 0$ inherently includes first-order terms.
>
> --------
>
> >### Summary
>
> We sincerely thank you for your insightful questions and approval of our motivation, novelty, and effectiveness.
>
> We sincerely hope all your concerns have been thoroughly addressed. We greatly **appreciate your support** and are more than happy to discuss further if there are any additional points we may have missed. Thank you!
>
> [1] Yushun Zhang, Congliang Chen, Naichen Shi, Ruoyu Sun, and Zhi-Quan Luo. Adam can converge without any modification on update rules. In NeurIPS, 2022.
>
> [2] Pang Wei Koh and Percy Liang. Understanding black-box predictions via influence functions. In ICML, volume 70 of Proceedings of Machine Learning Research, pp. 1885–1894. PMLR, 2017.
>
> [3] Barak A. Pearlmutter. Fast exact multiplication by the hessian. Neural Comput., 6(1):147–160, 1994.
>
> [4] Hangyu Wang, Jianghao Lin, Bo Chen, Yang Yang, Ruiming Tang, Weinan Zhang, and Yong Yu. Towards efficient and effective unlearning of large language models for recommendation. CoRR, abs/2403.03536, 2024.
>
> [5] Yulong Chen, Yang Liu, Liang Chen, and Yue Zhang. Dialogsum: A real-life scenario dialogue summarization dataset. In ACL/IJCNLP (Findings), volume ACL/IJCNLP 2021 of Findings of ACL, pp. 5062–5074. Association for Computational Linguistics, 2021.
>
> [6] Weihao Yu, Zhengyuan Yang, Linjie Li, Jianfeng Wang, Kevin Lin, Zicheng Liu, Xinchao Wang, and Lijuan Wang. Mm-vet: Evaluating large multimodal models for integrated capabilities. In ICML. OpenReview.net, 2024.

---

### Author Response · Authors · 2024-12-03
**General Response**

We sincerely appreciate the reviewers' efforts and valuable suggestions in reviewing our paper. We are pleased that all reviewers reached a positive consensus regarding the motivation, presentation, novelty, and experimental effectiveness of our work. In this response, we summarize the strengths of our work as recognized by the reviewers, highlight its key contributions, and detail the revisions and responses we have made to address all the concerns raised.

>### **Strengths**

- Motivation and Novelty: **novel instance-wise** unlearning method (`Reviewer 8gYQ`), address multiple unlearning scenarios (`Reviewer W27P`), addresses the complexity concerns (`Reviewer W27P`), **important and interesting** (`Reviewer 8gYQ`).

- Presentation: written clearly (`Reviewer nWJj`), easy to follow (`Reviewer 8gYQ`), with clearly presented formulations (`Reviewer nWJj`)

- Experimental effectiveness: **effectively** maintain good performance (`Reviewer 8XJc`), evaluations are **comprehensive** (`Reviewer 8gYQ`), **high accuracy** (`Reviewer 8gYQ`), **high efficiency** (`Reviewer 8XJc`), extensive experiments are conducted (`Reviewer nWJj`)

>### **Contributions of our work**

- **Unified Framework for Unlearning Tasks**: LLMEraser introduces a unified and efficient framework for parameter-efficient unlearning in LLMs. It generalizes three unlearning tasks—instance removal, query modification, and response correction—under a single, well-defined taxonomy.

- **Influence Function-Based Approach**: The framework leverages influence functions to approximate the impact of data points to be unlearned. This avoids the need for full model retraining or fine-tuning, significantly enhancing computational efficiency.

- **Efficiency Enhancements**: LLMEraser accelerates the computation of inverse Hessian-vector products required in influence functions by using fast Hessian-vector products and mini-batch processing. These optimizations make the framework suitable for large-scale LLM applications.

- **Comprehensive Evaluation and Practicality**: The proposed framework is demonstrated to perform effectively across diverse unlearning tasks. It also addresses practical privacy and security concerns in domain-specific fine-tuning scenarios.

>### **Responses**
- Reviewer ` W27P ` :
  - More experimental evaluation: We have supplemented the experimental results on generative tasks and included these results in Appendix G.
  - Evaluating efficiency and running-time complexity: We have added a theoretical analysis of the time and space complexity in Appendix F.
  - How adapters are used in Algorithm 1: We have revised the description of Algorithm 1 and added an introduction to the workflow of LLMEraser.

- Reviewer `8gYQ` :
  - Model performance on generative tasks: We have supplemented the experimental results on generative tasks and included these results in Appendix G.

- Reviewer `8XJc` :
  - Model performance on generative tasks: We have supplemented the experimental results on generative tasks and included these results in Appendix G.
  - Unlearning effectiveness: We have added explanations regarding unlearning effectiveness along with the corresponding experimental results.
  - Unlearning setting of data used in both pretraining and PEFT: We have added a discussion on scenarios where data is used simultaneously by both pretraining and PEFT, along with the corresponding experimental results.

- Reviewer `nWJj` :
  - Model performance on generative tasks: We have supplemented the experimental results on generative tasks and included these results in Appendix G.
  - Estimation errors analysis: We have included a theoretical analysis of approximation errors in Appendix E.
  - Discussion about memory efficiency: We have added a theoretical analysis of the time and space complexity in Appendix F.

We deeply appreciate the time and effort you have devoted to evaluating our work. Your insightful feedback has greatly motivated us to further improve and advance this research for the benefit of the broader academic community. If you have any additional suggestions or questions, we would be pleased to offer further explanations.

Best regards,

Authors of Paper 6232

---

### Meta-Review · Area_Chair_Mksy · 2024-12-19

**Metareview:**

The paper provides a unified view of different unlearning scenarios. The reviews agree that the approach provided by the authors is both novel and well motivated. The area of unlearning is interesting to many researchers, and the paper provides a practical solution. The reviews appreciated the deep analysis and promising empirical results. That said, the reviews raised concern that there is a need for more thorough experiments, and minor concerns about the clarity of the paper.

In the rebuttal, the authors added some experiments mitigating the concerns related to the empirical section and explained the vague or unclear parts of the paper. The provided material extends the existing content rather than requiring a new discussion, hence integrating it into the paper should be a simple task. Given this, the above mentioned strength and that the modified scores being unanimously towards accepting, I recommend accepting the paper, and urge the authors to carefully review the discussion and integrate the necessary parts into the final version of the paper.

**Additional Comments On Reviewer Discussion:**

The authors provided a convincing response in the rebuttal phase, leading to a consensus for the paper to be accepted.

---

### Decision · Program_Chairs · 2025-01-22

Accept (Poster)